# DEEP FRANK-WOLFE
# FOR NEURAL NETWORK OPTIMIZATION

**Leonard Berrada**[1]**, Andrew Zisserman**[1] **and M. Pawan Kumar**[1,2]
[1]Department of Engineering Science
  University of Oxford
[2]Alan Turing Institute
`{lberrada,az,pawan}@robots.ox.ac.uk`

## ABSTRACT

Learning a deep neural network requires solving a challenging optimization problem: it is a high-dimensional, non-convex and non-smooth minimization problem with a large number of terms. The current practice in neural network optimization is to rely on the stochastic gradient descent (SGD) algorithm or its adaptive variants. However, SGD requires a hand-designed schedule for the learning rate. In addition, its adaptive variants tend to produce solutions that generalize less well on unseen data than SGD with a hand-designed schedule. We present an optimization method that offers empirically the best of both worlds: our algorithm yields good generalization performance while requiring only one hyper-parameter. Our approach is based on a composite proximal framework, which exploits the compositional nature of deep neural networks and can leverage powerful convex optimization algorithms by design. Specifically, we employ the Frank-Wolfe (FW) algorithm for SVM, which computes an optimal step-size in closed-form at each time-step. We further show that the descent direction is given by a simple backward pass in the network, yielding the same computational cost per iteration as SGD. We present experiments on the CIFAR and SNLI data sets, where we demonstrate the significant superiority of our method over Adam, Adagrad, as well as the recently proposed BPGrad and AMSGrad. Furthermore, we compare our algorithm to SGD with a hand-designed learning rate schedule, and show that it provides similar generalization while often converging faster. The code is publicly available at `https://github.com/oval-group/dfw`.

## 1 INTRODUCTION

Since the introduction of back-propagation (Rumelhart et al., 1986), stochastic gradient descent (SGD) has been the most commonly used optimization algorithm for deep neural networks. While yielding remarkable performance on a variety of learning tasks, a downside of the SGD algorithm is that it requires a schedule for the decay of its learning rate. In the convex setting, curvature properties of the objective function can be used to design schedules that are hyper-parameter free and guaranteed to converge to the optimal solution (Bubeck, 2015). However, there is no analogous result of practical interest for the non-convex optimization problem of a deep neural network. An illustration of this issue is the diversity of learning rate schedules used to train deep convolutional networks with SGD: Simonyan & Zisserman (2015) and He et al. (2016) adapt the learning rate according to the validation performance, while Szegedy et al. (2015), Huang et al. (2017) and Loshchilov & Hutter (2017) use pre-determined schedules, which are respectively piecewise constant, geometrically decaying, and cyclic with a cosine annealing. While these protocols result in competitive or state-of-the-art results on their learning task, there does not seem to be a consistent methodology. As a result, finding such a schedule for a new setting is a time-consuming and computationally expensive effort.

To alleviate this issue, adaptive gradient methods have been developed (Zeiler, 2012, Kingma & Ba, 2015, Reddi et al., 2018), and borrowed from online convex optimization (Duchi et al., 2011). Typically, these methods only require the tuning of the initial learning rate, the other hyper-parameters being considered robust across applications. However, it has been shown that such adaptive gradient

methods obtain worse generalization than SGD (Wilson et al., 2017). This observation is corroborated by our experimental results.

In order to bridge this performance gap between existing adaptive methods and SGD, we introduce a new optimization algorithm, called Deep Frank-Wolfe (DFW). The DFW algorithm exploits the composite structure of deep neural networks to design an optimization algorithm that leverages efficient convex solvers. In more detail, we consider a composite (nested) optimization problem, with the loss as the outer function and the function encoded by the neural network as the inner one. At each iteration, we define a proximal problem with a first-order approximation of the neural network (linearized inner function), while keeping the loss function in its exact form (exact outer function). When the loss is the hinge loss, each proximal problem created by our formulation is exactly a linear SVM. This allows us to employ the powerful Frank-Wolfe (FW) algorithm as the workhorse of our procedure.

There are two by-design advantages to our method compared to the SGD algorithm. First, each iteration exploits more information about the learning objective, while preserving the same computational cost as SGD. Second, an optimal step-size is computed in closed-form by using the FW algorithm in the dual (Frank & Wolfe, 1956, Lacoste-Julien et al., 2013). Consequently, we do not need a hand-designed schedule for the learning rate. As a result, our algorithm is the first to provide competitive generalization error compared to SGD, all the while requiring a single hyper-parameter and often converging significantly faster.

We present two additional improvements to customize the use of the DFW algorithm to deep neural networks. First, we show how to smooth the loss function to avoid optimization difficulties arising from learning deep models with SVMs (Berrada et al., 2018). Second, we incorporate Nesterov momentum (Nesterov, 1983) to accelerate our algorithm.

We demonstrate the efficacy of our method on image classification with the CIFAR data sets (Krizhevsky, 2009) using two architectures: wide residual networks (Zagoruyko & Komodakis, 2016) and densely connected convolutional neural networks (Huang et al., 2017); we also provide experiments on natural language inference with a Bi-LSTM on the SNLI corpus (Bowman et al., 2015). We show that the DFW algorithm often strongly outperforms previous methods based on adaptive learning rates. Furthermore, it provides comparable or better accuracy to SGD with hand-designed learning rate schedules.

In conclusion, our contributions can be summed up as follows:

- We propose a proximal framework which preserves information from the loss function.
- For the first time for deep neural networks, we demonstrate how our formulation gives at each iteration (i) an optimal step-size in closed form and (ii) an update at the same computational cost as SGD.
- We design a novel smoothing scheme for the dual optimization of SVMs.
- To the best of our knowledge, the resulting DFW algorithm is the first to offer comparable or better generalization to SGD with a hand-designed schedule on the CIFAR data sets, all the while converging several times faster and requiring only a single hyperparameter.

## 2 RELATED WORK

**Non Gradient-Based Methods.** The success of a simple first-order method such as SGD has led to research in other more sophisticated techniques based on relaxations (Heinemann et al., 2016, Zhang et al., 2017a), learning theory (Goel et al., 2017), Bregman iterations (Taylor et al., 2016), and even second-order methods (Roux et al., 2008, Martens & Sutskever, 2012, Ollivier, 2013, Desjardins et al., 2015, Martens & Grosse, 2015, Grosse & Martens, 2016, Ba et al., 2017, Botev et al., 2017, Martens et al., 2018). While such methods hold a lot of promise, their relatively large per-iteration cost limits their scalability in practice. As a result, gradient-based methods continue to be the most popular optimization algorithms for learning deep neural networks.

**Adaptive Gradient Methods.** As mentioned earlier, one of the main challenges of using SGD is the design of a learning rate schedule. Several works proposed alternative first-order methods that

do not require such a schedule, by either modifying the descent direction or adaptively rescaling the step-size (Duchi et al., 2011, Zeiler, 2012, Schaul et al., 2013, Kingma & Ba, 2015, Zhang et al., 2017b, Reddi et al., 2018). However, as noted above, the adaptive variants of SGD sometimes provide subpar generalization (Wilson et al., 2017).

**Learning to Learn and Meta-Learning.**   Learning to learn approaches have also been proposed to optimize deep neural networks. Baydin et al. (2018) and Wu et al. (2018) learn the learning rate to avoid a hand-designed schedule and to improve practical performance. Such methods can be combined with our proposed algorithm to learn its proximal coefficient, instead of considering it as a fixed hyper-parameter to be tuned. Meta-learning approaches have also been suggested to learn the optimization algorithm (Andrychowicz et al., 2016, Ravi & Larochelle, 2017, Wichrowska et al., 2017, Li & Malik, 2017). This line of work, which is orthogonal to ours, could benefit from the use of DFW to optimize the meta-learner.

**Optimization and Generalization.**   Several works study the relationship between optimization and generalization in deep learning. In order to promote generalization within the optimization algorithm itself, Neyshabur et al. (2015; 2016) proposed the Path-SGD algorithm, which implicitly controls the capacity of the model. However, their method required the model to employ ReLU non-linearity only, which is an important restriction for practical purposes. Hardt et al. (2016), Arpit et al. (2017), Neyshabur et al. (2017), Hoffer et al. (2017) and Chaudhari & Soatto (2018) analyzed how existing optimization algorithms implicitly regularize deep neural networks. However this phenomenon is not yet fully understood, and the resulting empirical recommendations are sometimes opposing (Hardt et al., 2016, Hoffer et al., 2017).

**Proximal Methods.**   The back-propagation algorithm has been analyzed in a proximal framework in (Frerix et al., 2018). Yet, the resulting approach still requires the same hyper-parameters as SGD and incurs a higher computational cost per iteration.

**Linear SVM Sub-Problems.**   A main component of our formulation is to formulate sub-problems as linear SVMs. In an earlier work (Berrada et al., 2017), we showed that neural networks with piecewise linear activations could be trained with the CCCP algorithm (Yuille & Rangarajan, 2002), which yielded approximate SVM problems to be solved with the BCFW algorithm (Lacoste-Julien et al., 2013). However this algorithm only updates the parameters of one layer at a time, which slows down convergence significantly in practice. Closest to our approach are the works of (Hochreiter & Obermayer, 2005) and (Singh & Shawe-Taylor, 2018). Hochreiter & Obermayer (2005) suggested to create a local SVM based on a first-order Taylor expansion and a proximal term, in order to lower the error of every data sample while minimizing the changes in the weights. However their method operated in a non-stochastic setting, making the approach infeasible for large-scale data sets. Singh & Shawe-Taylor (2018), a parallel work to ours, also created an SVM problem using a first-order Taylor expansion, this time in a mini-batch setting. Their work provided interesting insights from a statistical learning theory perspective. While their method is well-grounded, its significantly higher cost per iteration impairs its practical speed and scalability. As such, it can be seen as complementary to our empirical work, which exploits a powerful solver and provides state-of-the-art scalability and performance.

## 3   PROBLEM FORMULATION

Before describing our formulation, we introduce some necessary notation. We use $\| \cdot \|$ to denote the Euclidean norm. Given a function $\phi$, $\partial\phi(\mathbf{u})\big|_{\hat{\mathbf{u}}}$ is the derivative of $\phi$ with respect to $\mathbf{u}$ evaluated at $\hat{\mathbf{u}}$. According to the situation, this derivative can be a gradient, a Jacobian or even a directional derivative. Its exact nature will be clear from context throughout the paper. We also introduce the first-order Taylor expansion of $\phi$ around the point $\hat{\mathbf{u}}$: $\mathcal{T}_{\hat{\mathbf{u}}}\phi(\mathbf{u}) = \phi(\hat{\mathbf{u}}) + (\partial\phi(\mathbf{u})\big|_{\hat{\mathbf{u}}})^\top (\mathbf{u} - \hat{\mathbf{u}})$. For a positive integer $p$, we denote the set $\{1, 2, ..., p\}$ as $[p]$. For simplicity, we assume that stochastic algorithms process only one sample at each iteration, although the methods can be trivially extended to mini-batches of size larger than one.

## 3.1 Learning Objective

We suppose we are given a data set $(\mathbf{x}_i, y_i)_{i \in [N]}$, where each $\mathbf{x}_i \in \mathbb{R}^d$ is a sample annotated with a label $y_i$ from the output space $\mathcal{Y}$. The data set is used to estimate a parameterized model represented by the function $\mathbf{f}$. Given its (flattened) parameters $\mathbf{w} \in \mathbb{R}^p$, and an input $\mathbf{x}_i \in \mathbb{R}^d$, the model predicts $\mathbf{f}(\mathbf{w}, \mathbf{x}_i) \in \mathbb{R}^{|\mathcal{Y}|}$, a vector with one score per element of the output space $\mathcal{Y}$. For instance, $\mathbf{f}$ can be a linear map or a deep neural network. Given a vector of scores per label $\mathbf{s} \in \mathbb{R}^{|\mathcal{Y}|}$, we denote by $\mathcal{L}(\mathbf{s}, y_i)$ the loss function that computes the risk of the prediction scores $\mathbf{s}$ given the ground truth label $y_i$. For example, the loss $\mathcal{L}$ can be cross-entropy or the multi-class hinge loss:

$$\text{(Cross-Entropy Loss)} \quad \mathcal{L}_{CE} : (\mathbf{s}, y) \in \mathbb{R}^{|\mathcal{Y}|} \times \mathcal{Y} \mapsto \log\left(\sum_{k \in \mathcal{Y}} \exp(s_k)\right) - s_y, \tag{1}$$

$$\text{(Multi-Class Hinge Loss)} \quad \mathcal{L}_{hinge} : (\mathbf{s}, y) \in \mathbb{R}^{|\mathcal{Y}|} \times \mathcal{Y} \mapsto \max\left\{\max_{k \in \mathcal{Y} \setminus \{y\}} \{s_k + 1 - s_y\}, 0\right\}. \tag{2}$$

The cross-entropy loss (1) tries to match the empirical distribution by driving incorrect scores as far as possible from the ground truth one. The hinge loss (2) attempts to create a minimal margin of one between correct and incorrect scores. The hinge loss has been shown to be more robust to over-fitting than cross-entropy, when combined with smoothing techniques that are common in the optimization literature (Berrada et al., 2018). To simplify notation, we introduce $\mathbf{f}_i(\mathbf{w}) = \mathbf{f}(\mathbf{w}, \mathbf{x}_i)$ and $\mathcal{L}_i(\mathbf{s}) = \mathcal{L}(\mathbf{s}, y_i)$ for each $i \in [N]$. Finally, we denote by $\rho(\mathbf{w})$ the regularization (typically the squared Euclidean norm). We now write the learning problem under its empirical risk minimization form:

$$\min_{\mathbf{w} \in \mathbb{R}^p} \rho(\mathbf{w}) + \frac{1}{N} \sum_{i \in [N]} \mathcal{L}_i(\mathbf{f}_i(\mathbf{w})). \tag{3}$$

## 3.2 A Proximal Approach

Our main contribution is a formulation which exploits the composite nature of deep neural networks in order to obtain a better approximation of the objective at each iteration. Thanks to the careful approximation design, this approach yields sub-problems that are amenable to efficient optimization by powerful convex solvers. In order to understand the intuition of our approach, we first present a proximal gradient perspective on SGD.

**The SGD Algorithm.** At iteration $t$, the SGD algorithm selects a sample $j$ at random and observes the objective estimate $\rho(\mathbf{w}_t) + \mathcal{L}_j(\mathbf{f}_j(\mathbf{w}_t))$. Then, given the learning rate $\eta_t$, it performs the following update on the parameters:

$$\mathbf{w}_{t+1} = \mathbf{w}_t - \eta_t \left( \partial\rho(\mathbf{w})\big|_{\mathbf{w}_t} + \partial\mathcal{L}_j(\mathbf{f}_j(\mathbf{w}))\big|_{\mathbf{w}_t} \right). \tag{4}$$

Equation (4) is the closed-form solution of a proximal problem where the objective has been linearized by the first-order Taylor expansion $\mathcal{T}_{\mathbf{w}_t}$ (Bubeck, 2015):

$$\mathbf{w}_{t+1} = \arg\min_{\mathbf{w} \in \mathbb{R}^p} \left\{ \frac{1}{2\eta_t} \|\mathbf{w} - \mathbf{w}_t\|^2 + \mathcal{T}_{\mathbf{w}_t}\rho(\mathbf{w}) + \mathcal{T}_{\mathbf{w}_t}[\mathcal{L}_j(\mathbf{f}_j(\mathbf{w}))] \right\}. \tag{5}$$

To see the relationship between (4) and (5), one can set the gradient with respect to $\mathbf{w}$ to 0 in equation (5), and observe that the resulting equation is exactly (4). In other words, SGD minimizes a first-order approximation of the objective, while encouraging proximity to the current estimate $\mathbf{w}_t$.

However, one can also choose to linearize only a part of the composite objective (Lewis & Wright, 2016). Choosing which part to approximate is a crucial decision, because it yields optimization problems with widely different properties. In this work, we suggest an approach that lends itself to fast optimization with robust convex solvers and preserves information about the learning task by keeping an exact loss function.

**Loss-Preserving Linearization.** In detail, at iteration $t$, with selected sample $j$, we introduce the proximal problem that linearizes the regularization $\rho$ and the model $\mathbf{f}_j$, but not the loss function $\mathcal{L}$:

$$\min_{\mathbf{w} \in \mathbb{R}^p} \left\{ \frac{1}{2\eta_t} \|\mathbf{w} - \mathbf{w}_t\|^2 + \mathcal{T}_{\mathbf{w}_t}\rho(\mathbf{w}) + \mathcal{L}_j(\mathcal{T}_{\mathbf{w}_t}\mathbf{f}_j(\mathbf{w})) \right\}. \tag{6}$$

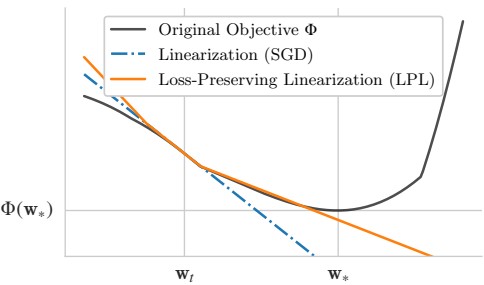 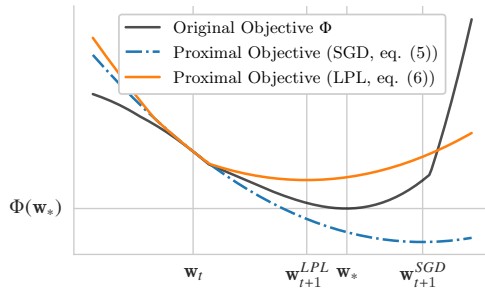

Figure 1: *We illustrate the different approximations on a synthetic composite objective function* $\Phi(\mathbf{w}) = \mathcal{L}(\mathbf{f}(\mathbf{w}))$ *($\Phi$ is plotted in black). In this example, $\mathcal{L}$ is a maximum of linear functions (similarly to a hinge loss) and $\mathbf{f}$ is a non-linear smooth map. We denote the current iterate by $\mathbf{w}_t$, and the point minimizing $\Phi$ by $\mathbf{w}_*$. On the left-hand side, one can observe how the SGD approximation is a single line (tangent at $\Phi(\mathbf{w}_t)$, in blue), while the LPL approximation is piecewise linear (in orange), and thus matches the objective curve (in black) more closely. On the right-hand side, an identical proximal term is added to both approximations to visualize equations (5) and (6). Thanks to the better accuracy of the LPL approximation, the iterate $\mathbf{w}_{t+1}^{LPL}$ gets closer to the solution $\mathbf{w}_*$ than $\mathbf{w}_{t+1}^{SGD}$. This effect is particularly true when the proximal coefficient $\frac{1}{2\eta_t}$ is small, or equivalently, when the learning rate $\eta_t$ is large. Indeed, the accuracy of the local approximation becomes more important when the proximal term is contributing less (e.g. when $\eta_t$ is large).*

In figure 1, we provide a visual comparison of equations (5) and (6) in the case of a piecewise linear loss. As will be seen, by preserving the loss function, we will be able to achieve good performance across a number of tasks with a fixed $\eta_t = \eta$. Consequently, we will provide the first algorithm to accurately learn deep neural networks with only a single hyper-parameter while offering similar performance compared to SGD with a hand-designed schedule.

## 4 THE DEEP FRANK-WOLFE ALGORITHM

### 4.1 ALGORITHM

We focus on the optimization of equation (6) when $\mathcal{L}$ is a multi-class hinge loss (2). The results of this section were originally derived for linear models (Lacoste-Julien et al., 2013). Our contribution is to show for the first time how they can be exploited for deep neural networks thanks to our formulation (6). We will refer to the resulting algorithm for neural networks as Deep Frank-Wolfe (DFW). We begin by stating the key advantage of our method.

**Proposition 1** (Optimal step-size, (Lacoste-Julien et al., 2013)). *Problem (6) with a hinge loss is amenable to optimization with Frank-Wolfe in the dual, which yields an optimal step-size $\gamma_t \in [0, 1]$ in closed-form at each iteration $t$.*

This optimal step-size can be obtained in closed-form because the hinge loss is convex and piecewise linear. In fact, the approach presented here can be applied to any loss function $\mathcal{L}$ that is convex and piecewise linear (another example would be the $l_1$ distance for regression for instance).

Since the step-size can be computed in closed-form, the main computational challenge is to obtain the update direction, that is, the conditional gradient of the dual. In the following result, we show that by taking a single step per proximal problem, this dual conditional gradient can be computed at the same cost as a standard stochastic gradient. The proof is available in appendix A.5.

**Proposition 2** (Cost per iteration). *If a single step is performed on the dual of (6), its conditional gradient is given by $-\partial \left( \rho(\mathbf{w}) + \mathcal{L}_y(\mathbf{f_x}(\mathbf{w})) \right) \big|_{\mathbf{w}_t}$. Given the step-size $\gamma_t$, the resulting update can be written as:*

$$\mathbf{w}_{t+1} = \mathbf{w}_t - \eta \left[ \partial \rho(\mathbf{w}) \big|_{\mathbf{w}_t} + \gamma_t \partial \mathcal{L}_j(\mathbf{f}_j(\mathbf{w})) \big|_{\mathbf{w}_t} \right] \tag{7}$$

In other words, the cost per iteration of the DFW algorithm is the same as SGD, since the update only requires standard stochastic gradients. In addition, we point out that in a mini-batch setting, the conditional gradient is given by the average of the gradients over the mini-batch. As a consequence, we can use batch Frank-Wolfe in the dual rather than coordinate-wise updates, with the same parallelism as SGD over the samples of a mini-batch.

One can observe how the update (7) exploits the optimal step-size $\gamma_t \in [0, 1]$ given by Proposition 1. There is a geometric interpretation to the role of this step-size $\gamma_t$. When $\gamma_t$ is set to its minimal value 0, the resulting iterate does not move along the direction $\partial \mathcal{L}_j(\mathbf{f}_j(\mathbf{w}))\big|_{\mathbf{w}_t}$. Since the step-size is optimal, this can only happen if the current iterate is detected to be at a minimum of the piecewise linear approximation. Conversely, when $\gamma_t$ reaches its maximal value 1, the algorithm tries to move as far as possible along the direction $\partial \mathcal{L}_j(\mathbf{f}_j(\mathbf{w}))\big|_{\mathbf{w}_t}$. In that case, the update is the same as the one obtained by SGD (as given by equation (4)). In other words, $\gamma_t$ can automatically decay the effective learning rate, hereby preventing the need to design a learning rate schedule by hand.

As mentioned previously, the DFW algorithm performs only one step per proximal problem. Since problem (6) is only an approximation of the original problem (3), it may be unnecessarily expensive to solve it very accurately. Therefore taking a single step per proximal problem may help the DFW algorithm to converge faster. This is confirmed by our experimental results, which show that DFW is often able to minimize the learning objective (3) at greater speed than SGD.

## 4.2 IMPROVEMENTS FOR DEEP NEURAL NETWORKS

We present two improvements to customize the application of our algorithm to deep neural networks.

**Smoothing.** The SVM loss is non-smooth and has sparse derivatives, which can cause difficulties when training a deep neural network (Berrada et al., 2018). In Appendix A.6, we derive a novel result that shows how we can exploit the smooth primal cross-entropy direction and inexpensively detect when to switch back to using the standard conditional gradient.

**Nesterov Momentum.** To take advantage of acceleration similarly to the SGD baseline, we adapt the Nesterov momentum to the DFW algorithm. We defer the details to the appendix in A.7 for space reasons. We further note that the momentum coefficient $\mu$ is typically set to a high value, say 0.9, and does not contribute significantly to the computational cost of cross-validation.

## 4.3 ALGORITHM SUMMARY

The main steps of DFW are shown in Algorithm 1. As the key feature of our approach, note that the step-size is computed in closed-form in step 10 of the algorithm (colored in blue).

Note that only the hyper-parameter $\eta$ will be tuned in our experiments: we will use the same batch-size, momentum and number of epochs as the baselines in our experiments (unless specified otherwise). In addition, we point out again that when $\gamma_t = 1$, we recover the SGD step with Nesterov momentum.

In sections A.5 and A.6 of the appendix, we detail the derivation of the optimal step-size (step 10) and the computation of the search direction (step 7). The computation of the dual search direction is omitted here for space reasons. However, its implementation is straightforward in practice, and its computational cost is linear in the size of the output space.

Finally, we emphasize that the DFW algorithm is motivated by an empirical perspective. While our method is not guaranteed to converge, our experiments show an effective minimization of the learning objective for the problems encountered in practice.

## 5 EXPERIMENTS

We compare the Deep Frank Wolfe (DFW) algorithm to the state-of-the-art optimizers. We show that, across diverse data sets and architectures, the DFW algorithm outperforms adaptive gradient methods (with the exception of one setting, DN-10, where it obtains similar performance to AMSGrad and BPGrad). In addition, the DFW algorithm offers competitive and sometimes superior performance to

---

**Algorithm 1** *The Deep Frank-Wolfe Algorithm*

---

**Require:** proximal coefficient $\eta$, initial point $\mathbf{w}_0 \in \mathbb{R}^p$, momentum coefficient $\mu$, number of epochs

1: $t = 0$
2: $\mathbf{z}_0 = 0$                             $\triangleright$ Momentum velocity (initialization)
3: **for** each epoch **do**
4:      **for** each mini-batch $\mathcal{B}$ **do**
5:          Receive data of mini-batch $(\mathbf{x}_i, y_i)_{i \in \mathcal{B}}$
6:          $\forall i \in \mathcal{B}, \ \mathbf{b}_t^{(i)}(\mathbf{w}_t) = (f_{\mathbf{x}_i, \bar{y}}(\mathbf{w}_t) - f_{\mathbf{x}_i, y_i}(\mathbf{w}_t) + \Delta(\bar{y}, y_i))_{\bar{y} \in \mathcal{Y}}$      $\triangleright$ Forward pass
7:          $\forall i \in \mathcal{B}, \ \mathbf{s}_t^{(i)} \leftarrow \texttt{get\_s}(\mathbf{b}_t^{(i)}(\mathbf{w}_t))$      $\triangleright$ Dual direction (details in Appendix A.6)
8:          $\boldsymbol{\delta}_t = \partial \left( \frac{1}{|\mathcal{B}|} \sum_{i \in \mathcal{B}} (\mathbf{s}_t^{(i)})^\top \mathbf{b}_t^{(i)}(\mathbf{w}) \right) \big|_{\mathbf{w}_t}$      $\triangleright$ Derivative of (smoothed) loss function
9:          $\mathbf{r}_t = \partial \rho(\mathbf{w}) \big|_{\mathbf{w}_t}$      $\triangleright$ Derivative of regularization
10:          $\gamma_t = (-\eta \boldsymbol{\delta}_t^\top \mathbf{r}_t + \frac{1}{|\mathcal{B}|} \sum_{i \in \mathcal{B}} (\mathbf{s}_t^{(i)})^\top \mathbf{b}_t^{(i)}(\mathbf{w}_t) / (\eta \|\boldsymbol{\delta}_t\|^2)$ clipped to $[0, 1]$      $\triangleright$ Step-size
11:          $\mathbf{z}_{t+1} = \mu \mathbf{z}_t - \eta \gamma_t (\mathbf{r}_t + \boldsymbol{\delta}_t)$      $\triangleright$ Velocity accumulation
12:          $\mathbf{w}_{t+1} = \mathbf{w}_t - \eta [\mathbf{r}_t + \gamma_t \boldsymbol{\delta}_t] + \mu \mathbf{z}_{t+1}$      $\triangleright$ Parameters update
13:          $t = t + 1$
14:      **end for**
15: **end for**

---

SGD at a lower computational cost, even though SGD has the advantage of a hand-designed schedule that has been chosen separately for each of these tasks.

Our experiments are implemented in pytorch (Paszke et al., 2017), and the code is available at `https://github.com/oval-group/dfw`. All models are trained on a single Nvidia Titan Xp card.

### 5.1 IMAGE CLASSIFICATION WITH CONVOLUTIONAL NEURAL NETWORKS

**Data Set & Architectures.** The CIFAR-10/100 data sets contain 60,000 RGB natural images of size $32 \times 32$ with 10/100 classes (Krizhevsky, 2009). We split the training set into 45,000 training samples and 5,000 validation samples, and use 10,000 samples for testing. The images are centered and normalized per channel. Unless specified otherwise, no data augmentation is employed. We perform our experiments on two modern architectures of deep convolutional neural networks: wide residual networks (Zagoruyko & Komodakis, 2016), and densely connected convolutional networks (Huang et al., 2017). Specifically, we employ a wide residual network of depth 40 and width factor 4, which has 8.9M parameters, and a "bottleneck" densely connected convolutional neural network of depth 40 and growth factor 40, which has 1.9M parameters. We refer to these architectures as WRN and DN respectively. All the following experimental details follow the protocol of (Zagoruyko & Komodakis, 2016) and (Huang et al., 2017). The only difference is that, instead of using 50,000 samples for training, we use 45,000 samples for training, and 5,000 samples for the validation set, which we found to be essential for all adaptive methods. While Deep Frank Wolfe (DFW) uses an SVM loss, the baselines are trained with the Cross-Entropy (CE) loss since this resulted in better performance.

**Method.** We compare DFW to the most common adaptive learning rates currently used: Adagrad (Duchi et al., 2011), Adam (Kingma & Ba, 2015), the corrected version of Adam called AMSGrad (Reddi et al., 2018), and BPGrad (Zhang et al., 2017b). For these methods and for DFW, we cross-validate the initial learning rate as a power of 10. We also evaluate the performance of SGD with momentum (simply referred to as SGD), for which we follow the protocol of (Zagoruyko & Komodakis, 2016) and (Huang et al., 2017). For all methods, we set a budget of 200 epochs for WRN and 300 epochs for DN. Furthermore, the batch-size is respectively set to 128 and 64 for WRN and DN as in (Zagoruyko & Komodakis, 2016) and (Huang et al., 2017). For DN, the $l_2$ regularization is set to $10^{-4}$ as in (Huang et al., 2017). For WRN, the $l_2$ is cross-validated between $5.10^{-4}$, as in (Zagoruyko & Komodakis, 2016), and $10^{-4}$, a more usual value that we have found to perform better for some of the methods (in particular DFW, since the corresponding loss function is an SVM instead of CE, for which the value of $5.10^{-4}$ was designed). The value of the Nesterov momentum is set to

0.9 for BPGrad, SGD and DFW. DFW has only one hyper-parameter to tune, namely $\eta$, which is analogous to an initial learning rate. For SGD, the initial learning rate is set to 0.1 on both WRN and DN. Following (Zagoruyko & Komodakis, 2016) and (Huang et al., 2017), it is then divided by 5 at epochs 60, 120 and 180 for WRN, and by 10 at epochs 150 and 225 for DN.

**Results.** We present the results in Table 1.

| Architecture | Optimizer | CIFAR-10 Test Accuracy (%) | CIFAR-100 Test Accuracy (%) |
|---|---|---|---|
| WRN | Adagrad | 86.07 | 57.64 |
| | Adam | 84.86 | 58.46 |
| | AMSGrad | 86.08 | 60.73 |
| | BPGrad | 88.62 | 60.31 |
| | DFW | **90.18** | **67.83** |
| | SGD | 90.08 | 66.78 |
| DN | Adagrad | 87.32 | 56.47 |
| | Adam | 88.44 | 64.61 |
| | AMSGrad | 90.53 | 68.32 |
| | BPGrad | **90.85** | 59.36 |
| | DFW | 90.22 | **69.55** |
| | SGD | **92.02** | **70.33** |

Table 1: *Results on the CIFAR data sets without data augmentation. In black, all adaptive methods have a single hyper-parameter for their step-size. In red, SGD benefits from a hand-designed schedule. DFW outperforms all baselines on the WRN architecture, by a margin of 7% for adaptive gradient methods on CIFAR-100. On the DN-100 task, DFW exceeds the accuracy of Adagrad by 14%.*

Observe that DFW significantly outperforms the adaptive gradient methods, particularly on the more challenging CIFAR-100 data set. On the WRN-CIFAR-100 task in particular, DFW obtains a testing accuracy which is about 7% higher than all other adaptive methods and outperforms SGD with a hand-designed schedule by 1%. The inferior generalization of adaptive gradient methods is consistent with the findings of Wilson et al. (2017). On all tasks, the accuracy of DFW is comparable to SGD. Note that DFW converges significantly faster than SGD: the network reaches its final performance several times faster than SGD in all cases. We illustrate this with an example in figure 2, which plots the training and validation errors on DN-CIFAR-100. In figure 3, one can see how the step-size is automatically decayed by DFW on this same experiment: we compare the effective step-size $\gamma_t \eta$ for DFW to the manually tuned $\eta_t$ for SGD.

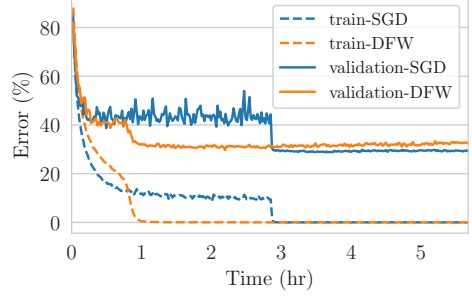

Figure 2: *Training and validation error during the training of DN on CIFAR-100. DFW converges significantly faster than SGD.*

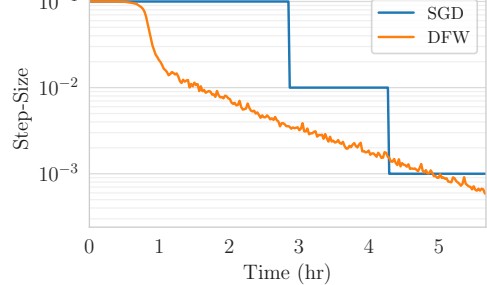

Figure 3: *The (automatic) evolution of $\gamma_t \eta$ for the DFW algorithm compared to the "staircase" hand-designed schedule of $\eta_t$ for SGD.*

**Data Augmentation.** Since data augmentation provides a significant boost to the final accuracy, we provide additional results that make use of it. Specifically, we randomly flip the images horizontally and randomly crop them with four pixels padding. For methods that do not use a hand-designed

schedule, such data augmentation introduces additional variance which makes the adaptation of the step-size more difficult. Therefore we allow the batch size of adaptive methods (e.g. all methods but SGD) to be chosen as 1x, 2x or 4x, where x is the original value of batch-size (64 for DN, 128 for WRN). Due to the heavy computational cost of the cross-validation (we tune the batch-size, regularization and initial learning rate), we provide results for SGD, DFW and the best performing adaptive gradient method, which is AMSGrad. For SGD the hyper-parameters are kept the same as in (Zagoruyko & Komodakis, 2016) and (Huang et al., 2017). We present the results in Table 2.

| Architecture | Optimizer | CIFAR-10 Test Accuracy (%) | CIFAR-100 Test Accuracy (%) |
|---|---|---|---|
| WRN | AMSGrad | 90.06 | 67.75 |
| | DFW | **94.71** | **74.71** |
| | SGD | **95.40** | **77.78** |
| | SGD* | **95.47** | **78.82** |
| DN | AMSGrad | 91.78 | 69.58 |
| | DFW | **94.88** | **73.20** |
| | SGD | **95.26** | **76.26** |

Table 2: *Results on the CIFAR data sets with data augmentation. In black, all adaptive methods have a single hyper-parameter for their step-size. In red, SGD benefits from a hand-designed schedule. On the fourth line, SGD\* refers to the result reported in Table 5 of (Huang et al., 2017). The small difference between the results of SGD and SGD\* can be explained by the fact that we use 5,000 fewer training samples in our experiments (these are kept for validation). The results of this table show that DFW systematically outperforms AMSGrad on this task (by up to 7% on WRN-100).*

These results confirm that DFW consistently outperforms AMSGrad, which is the best adaptive baseline on these tasks. In particular, DFW obtains a test accuracy which is 7% better than AMSGrad on WRN-100.

## 5.2 NATURAL LANGUAGE INFERENCE WITH RECURRENT NEURAL NETWORKS

**Data Set.** The Stanford Natural Language Inference (SNLI) data set is a large corpus of 570k pairs of sentences (Bowman et al., 2015). Each sentence is labeled by one of the three possible labels: entailment, neutral and contradiction. This allows the model to learn the semantics of the text data from a three-way classification problem. Thanks to its scale and its supervised labels, this data set allows large neural networks to learn high-quality text embeddings. As Conneau et al. (2017) demonstrate, the SNLI corpus can thus be used as a basis for transfer learning in natural language processing, in the same way that the ImageNet data set is used for pre-training in computer vision.

**Method.** We follow the protocol of (Conneau et al., 2017) to learn their best model, namely a bi-directional LSTM of about 47M parameters. In particular, the reported results use SGD with an initial learning rate of 0.1 and a hand-designed schedule that adapts to the variations of the validation set: if the validation accuracy does not improve, the learning rate is divided by a factor of 5. We also report results on Adagrad, Adam, AMSGrad and BPGrad. Following the official SGD baseline, Nesterov momentum is deactivated. Using their open-source implementation, we replace the optimization by the DFW algorithm, the CE loss by an SVM, and leave all other components unchanged. In this experiment, we use the conditional gradient direction rather than the CE gradient, since three-way classification does not cause sparsity in the derivative of the hinge loss (which is the issue that originally motivated our use of a different direction). We cross-validate our initial proximal term as a power of ten, and do not manually tune any schedule. In order to disentangle the importance of the loss function from the optimization algorithm, we run the baselines with both an SVM loss and a CE loss. The initial learning rate of the baselines is also cross-validated as a power of ten.

**Results.** The results are presented in Table 3.

| Optimizer | Loss | Adagrad | Adam | AMSGrad | BPGrad | DFW | SGD | SGD* |
|---|---|---|---|---|---|---|---|---|
| Test Accuracy (%) | CE | 83.8 | 84.5 | 84.2 | 83.6 | - | 84.7 | 84.5 |
| | SVM | 84.6 | 85.0 | 85.1 | 84.2 | **85.2** | **85.2** | - |

Table 3: *Results on the Stanford Natural Language Inference corpus. In black, all adaptive methods have a single hyper-parameter for their step-size. In red, SGD benefits from a hand-designed schedule. SGD* refers to the result reported in (Conneau et al., 2017). The other results have been obtained with their open-source implementation in our own experiments.*

Note that these results outperform the reported testing accuracy of 84.5% in (Conneau et al., 2017) that is obtained with CE. This experiment, which is performed on a completely different architecture and data set than the previous one, confirms that DFW outperforms adaptive gradient methods and matches the performance of SGD with a hand-designed learning rate schedule.

## 6 THE IMPORTANCE OF THE STEP-SIZE

### 6.1 IMPACT ON GENERALIZATION

It is worth discussing the subtle relationship between optimization and generalization. In order to emphasize the impact of implicit regularization, all results presented in this section do not use data augmentation. As a first illustrative example, we consider the following experiment: we take the protocol to train the DN network on CIFAR-100 with SGD, and simply change the initial learning rate to be ten times smaller, and the budget of epochs to be ten times larger. As a result, the final training objective significantly decreases from 0.33 to 0.069. Yet at the same time, the best validation accuracy decreases from 70.94% to 68.7%. A similar effect occurs when decreasing the value of the momentum, and we have observed this across various convolutional architectures. In other words, accurate optimization is less important for generalization than the implicit regularization of a high learning rate.

We have observed DFW to accurately optimize the learning objective in our experiments. However, given the above observation, we believe that its good generalization properties are rather due to its capability to usually maintain a high learning rate at an early stage. Similarly, the good generalization performance of SGD may be due to its schedule with a large number of steps at a high learning rate.

### 6.2 SENSITIVITY ANALYSIS

The previous section has qualitatively hinted at the importance of the step-size for generalization. Here we quantitatively analyze the impact of the initial learning rate $\eta$ on both the training accuracy (quality of optimization) and the validation accuracy (quality of generalization). We compare results of the DFW and SGD algorithms on the CIFAR data sets when varying the value of $\eta$ as a power of 10. The results on the validation set are summarized in figure 4, and the performance on the training set is reported in Appendix B.

On the training set, both methods obtain nearly perfect accuracy across at least three orders of magnitude of $\eta$ (details in Appendix B.4). In contrast, the results of figure 4 confirm that the validation performance is sensitive to the choice of $\eta$ for both methods.

In some cases where $\eta$ is high, SGD obtains a better performance than DFW. This is because the hand-designed schedule of SGD enforces a decay of $\eta$, while the DFW algorithm relies on an automatic decay of the step-size $\gamma_t$ for effective convergence. This automatic decay may not happen if a small proximal term (large $\eta$) is combined with a local approximation that is not sufficiently accurate (for instance with a small batch-size).

However, if we allow the DFW algorithm to use a larger batch size, then the local approximation becomes more accurate and it can handle large values of $\eta$ as well. Interestingly, choosing a larger batch-size and a larger value of $\eta$ can result in better generalization. For instance, by using a batch-size of 256 (instead of 64) and $\eta = 1$, DFW obtains a test accuracy of 72.64% on CIFAR-100 with the DN architecture (SGD obtains 70.33% with the settings of (Huang et al., 2017)).

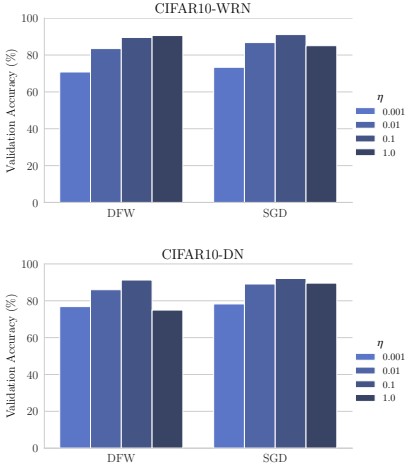
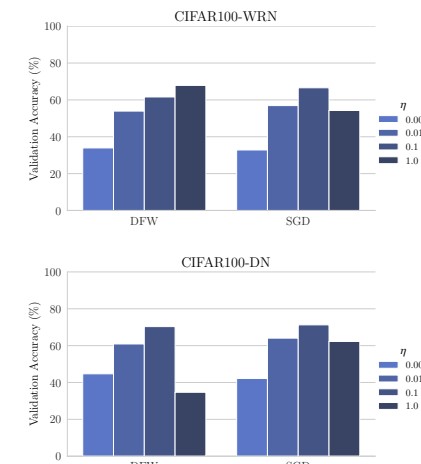

Figure 4: *Visualization of the sensitivity analysis for the choice of initial learning rate η on the CIFAR data sets. Each subplot displays the best validation accuracy for DFW and SGD. Similar plots are available in larger format in Appendix B.4.*

### 6.3 DISCUSSION

Our empirical evidence indicates that the initial learning rate can be a crucial hyper-parameter for good generalization. We have observed in our experiments that such a choice of high learning rate provides a consistent improvement for convolutional neural networks: accurate minimization of the training objective with large initial steps usually leads to good generalization. Furthermore, as mentioned in the previous section, it is sometimes beneficial to even increase the batch-size in order to be able to train the model using large initial steps.

In the case of recurrent neural networks, however, this effect is not as distinct. Additional experiments on different recurrent architectures have showed variations in the impact of the learning rate and in the best-performing optimizer. Further analysis would be required to understand the effects at play.

## 7 CONCLUSION

We have introduced DFW, an efficient algorithm to train deep neural networks. DFW predominantly outperforms adaptive gradient methods, and obtains similar performance to SGD without requiring a hand-designed learning rate schedule.

We emphasize the generality of our framework in Section 3, which enables the training of deep neural networks to benefit from any advance on optimization algorithms for linear SVMs. This framework could also be applied to other loss functions that yield efficiently solvable proximal problems. In particular, our algorithm already supports the use of structured prediction loss functions (Taskar et al., 2003, Tsochantaridis et al., 2004), which can be used, for instance, for image segmentation.

We have mentioned the intricate relationship between optimization and generalization in deep learning. This illustrates a major difficulty in the design of effective optimization algorithms for deep neural networks: the learning objective does not include all the regularization needed for good generalization. We believe that in order to further advance optimization for deep neural networks, it is essential to alleviate this problem and expose a clear objective function to optimize.

#### ACKNOWLEDGMENTS

This work was supported by the EPSRC grants AIMS CDT EP/L015987/1, Seebibyte EP/M013774/1, EP/P020658/1 and TU/B/000048, and by Yougov. We also thank the Nvidia Corporation for the GPU donation.

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

## A    PROOFS & ALGORITHMS

For completeness, we prove results for our specific instance of Structural SVM problem. We point out that the proofs of sections A.1, A.2 and A.3 are adaptations from (Lacoste-Julien et al., 2013). Propositions are numbered according to their appearance in the paper.

### A.1    PRELIMINARIES

In this section, we assume the loss $\mathcal{L}$ to be a hinge loss:

$$\mathcal{L}_{hinge} : (\mathbf{u}, y) \in \mathbb{R}^{|\mathcal{Y}|} \times \mathcal{Y} \mapsto \max \left\{ \max_{\bar{y} \in \mathcal{Y} \setminus \{y\}} \{u_{\bar{y}} + 1 - u_y\}, 0 \right\} \tag{8}$$

We suppose that we have received a sample $(\mathbf{x}, y)$. We simplify the notation $\mathbf{f}(\mathbf{w}, \mathbf{x}) = \mathbf{f}_{\mathbf{x}}(\mathbf{w})$ and $\mathcal{L}(\mathbf{u}, y) = \mathcal{L}_y(\mathbf{u})$. For simplicity of the notation, and without loss of generality, we consider the proximal problem obtained at time $t = 0$:

$$\min_{\mathbf{w} \in \mathbb{R}^p} \left\{ \frac{1}{2\eta} \|\mathbf{w} - \mathbf{w}_0\|^2 + \mathcal{T}_{\mathbf{w}_0} \rho(\mathbf{w}) + \mathcal{L}_y \left( \mathcal{T}_{\mathbf{w}_0} \mathbf{f}_{\mathbf{x}}(\mathbf{w}) \right) \right\}. \tag{9}$$

Let us define the classification task loss:

$$\text{For } (\bar{y}, y) \in \mathcal{Y}^2, \Delta(\bar{y}, y) = \begin{cases} 0 & \text{if } \bar{y} = y, \\ 1 & \text{otherwise.} \end{cases} \tag{10}$$

Using this notation, the multi-class hinge loss can be written as:

$$\mathcal{L}_{hinge}(\mathbf{u}, y) = \max_{\bar{y} \in \mathcal{Y}} \{u_{\bar{y}} + \Delta(\bar{y}, y) - u_y\}. \tag{11}$$

Indeed, we can successively write:

$$\begin{aligned} \mathcal{L}_{hinge}(\mathbf{u}, y) &= \max \left\{ \max_{\bar{y} \in \mathcal{Y} \setminus \{y\}} \{u_{\bar{y}} + 1 - u_y\}, 0 \right\}, \\ &= \max_{\bar{y} \in \mathcal{Y} \setminus \{y\}} \{\max \{u_{\bar{y}} + 1 - u_y, 0\}\}, \\ &= \max_{\bar{y} \in \mathcal{Y} \setminus \{y\}} \{\max \{u_{\bar{y}} + \Delta(\bar{y}, y) - u_y, 0\}\}, \\ &= \max_{\bar{y} \in \mathcal{Y}} \{\max \{u_{\bar{y}} + \Delta(\bar{y}, y) - u_y, 0\}\}, \\ &= \max_{\bar{y} \in \mathcal{Y}} \{u_{\bar{y}} + \Delta(\bar{y}, y) - u_y\}. \end{aligned} \tag{12}$$

We are now going to re-write problem (9) as the sum of a quadratic term and a pointwise maximum of linear functions. For $\bar{y} \in \mathcal{Y}$, let us define:

$$\begin{aligned} \mathbf{a}_{\bar{y}} &= \partial \rho(\mathbf{w})\big|_{\mathbf{w}_0} + \partial f_{\mathbf{x}, \bar{y}}(\mathbf{w})\big|_{\mathbf{w}_0} - \partial f_{\mathbf{x}, y}(\mathbf{w})\big|_{\mathbf{w}_0}, \\ b_{\bar{y}} &= \rho(\mathbf{w}_0) + f_{\mathbf{x}, \bar{y}}(\mathbf{w}_0) - f_{\mathbf{x}, y}(\mathbf{w}_0) + \Delta(\bar{y}, y). \end{aligned} \tag{13}$$

Then we have that:

$$\begin{aligned} \max_{\bar{y} \in \mathcal{Y}} \left\{ \mathbf{a}_{\bar{y}}^{\top}(\mathbf{w} - \mathbf{w}_0) + b_{\bar{y}} \right\} &= \max_{\bar{y} \in \mathcal{Y}} \left\{ \left( \partial \rho(\mathbf{w})\big|_{\mathbf{w}_0} + \partial f_{\mathbf{x}, \bar{y}}(\mathbf{w})\big|_{\mathbf{w}_0} - \partial f_{\mathbf{x}, y}(\mathbf{w})\big|_{\mathbf{w}_0} \right)^{\top}(\mathbf{w} - \mathbf{w}_0) \right. \\ &\qquad \left. + \rho(\mathbf{w}_0) + f_{\mathbf{x}, \bar{y}}(\mathbf{w}_0) - f_{\mathbf{x}, y}(\mathbf{w}_0) + \Delta(\bar{y}, y) \right\}, \\ &= \rho(\mathbf{w}_0) + \partial \rho(\mathbf{w})\big|_{\mathbf{w}_0}^{\top}(\mathbf{w} - \mathbf{w}_0) \\ &\qquad + \max_{\bar{y} \in \mathcal{Y}} \left\{ \partial f_{\mathbf{x}, \bar{y}}(\mathbf{w})\big|_{\mathbf{w}_0}^{\top}(\mathbf{w} - \mathbf{w}_0) + f_{\mathbf{x}, \bar{y}}(\mathbf{w}_0) + \Delta(\bar{y}, y) \right\} \\ &\qquad - f_{\mathbf{x}, y}(\mathbf{w}_0) - \partial f_{\mathbf{x}, y}(\mathbf{w})\big|_{\mathbf{w}_0}^{\top}(\mathbf{w} - \mathbf{w}_0), \\ &= \mathcal{T}_{\mathbf{w}_0} \rho(\mathbf{w}) + \mathcal{L} \left( \mathcal{T}_{\mathbf{w}_0} \mathbf{f}_{\mathbf{x}}(\mathbf{w}), y \right). \end{aligned} \tag{14}$$

Therefore, problem (9) can be written as:

$$\min_{\mathbf{w}\in\mathbb{R}^p}\left\{\frac{1}{2\eta}\|\mathbf{w}-\mathbf{w}_0\|^2+\max_{\bar{y}\in\mathcal{Y}}\left\{\mathbf{a}_{\bar{y}}^{\top}(\mathbf{w}-\mathbf{w}_0)+b_{\bar{y}}\right\}\right\}. \tag{15}$$

We notice that the term $\rho(\mathbf{w}_0)$ in $\mathbf{b}$ is a constant that does not depend on $\mathbf{w}$ nor $\bar{y}$, therefore we can simplify the expression of $\mathbf{b}$ to:

$$b_{\bar{y}}=f_{\mathbf{x},\bar{y}}(\mathbf{w}_0)-f_{\mathbf{x},y}(\mathbf{w}_0)+\Delta(\bar{y},y). \tag{16}$$

We introduce the following notation:

$$\hat{\mathbf{w}}=\mathbf{w}-\mathbf{w}_0, \tag{17}$$

$$\mathcal{P}=\{\boldsymbol{\alpha}\in\mathbb{R}_+^{|\mathcal{Y}|}:\sum_{\bar{y}\in\mathcal{Y}}\alpha_{\bar{y}}=1\}, \tag{18}$$

$$A=(\eta\mathbf{a}_{\bar{y}})_{\bar{y}\in\mathcal{Y}}\in\mathbb{R}^{p\times|\mathcal{Y}|}. \tag{19}$$

We will also use the indicator vector: $\mathbb{1}_y\in\mathbb{R}^{|\mathcal{Y}|}$, which is equal to 1 at index $y$ and 0 elsewhere.

## A.2  DUAL OBJECTIVE

**Lemma 1** (Dual Objective). *The Lagrangian dual of (9) is given by:*

$$\max_{\boldsymbol{\alpha}\in\mathcal{P}}\left\{-\frac{1}{2\eta}\|A\boldsymbol{\alpha}\|^2+\mathbf{b}^{\top}\boldsymbol{\alpha}\right\}. \tag{20}$$

*Given the dual variables $\boldsymbol{\alpha}$, the primal can be computed as $\hat{\mathbf{w}}=-A\boldsymbol{\alpha}$.*

*Proof.* We derive the Lagrangian of the primal problem. For that, we write the problem in the following equivalent ways:

$$\min_{\hat{\mathbf{w}}\in\mathbb{R}^p}\left\{\frac{1}{2\eta}\|\hat{\mathbf{w}}\|^2+\max_{\bar{y}\in\mathcal{Y}}\left\{\mathbf{a}_{\bar{y}}^{\top}\hat{\mathbf{w}}+b_{\bar{y}}\right\}\right\}, \tag{21}$$

$$\min_{\substack{\hat{\mathbf{w}}\in\mathbb{R}^p\\\xi\in\mathbb{R}}}\left\{\frac{1}{2\eta}\|\hat{\mathbf{w}}\|^2+\xi\right\}\text{ subject to: }\forall\bar{y}\in\mathcal{Y},\ \mathbf{a}_{\bar{y}}^{\top}\hat{\mathbf{w}}+b_{\bar{y}}\leq\xi, \tag{22}$$

$$\min_{\substack{\hat{\mathbf{w}}\in\mathbb{R}^p\\\xi\in\mathbb{R}}}\sup_{\boldsymbol{\alpha}\geq 0}\left\{\frac{1}{2\eta}\|\hat{\mathbf{w}}\|^2+\xi+\sum_{\bar{y}\in\mathcal{Y}}\alpha_{\bar{y}}\left(\mathbf{a}_{\bar{y}}^{\top}\hat{\mathbf{w}}+b_{\bar{y}}-\xi\right)\right\}, \tag{23}$$

$$\sup_{\boldsymbol{\alpha}\geq 0}\min_{\substack{\hat{\mathbf{w}}\in\mathbb{R}^p\\\xi\in\mathbb{R}}}\underbrace{\left\{\frac{1}{2\eta}\|\hat{\mathbf{w}}\|^2+\xi+\sum_{\bar{y}\in\mathcal{Y}}\alpha_{\bar{y}}\left(\mathbf{a}_{\bar{y}}^{\top}\hat{\mathbf{w}}+b_{\bar{y}}-\xi\right)\right\}}_{\Lambda(\hat{\mathbf{w}},\xi,\boldsymbol{\alpha})}\quad\text{(by strong duality).} \tag{24}$$

We can now write the KKT conditions of the inner minimization problem:

$$\begin{aligned}\frac{\partial\Lambda(\hat{\mathbf{w}},\xi,\boldsymbol{\alpha})}{\partial\xi}&=0:\quad 1-\sum_{\bar{y}\in\mathcal{Y}}\alpha_{\bar{y}}=0,\\\frac{\partial\Lambda(\hat{\mathbf{w}},\xi,\boldsymbol{\alpha})}{\partial\hat{\mathbf{w}}}&=\mathbf{0}:\quad\frac{1}{\eta}\hat{\mathbf{w}}+\sum_{\bar{y}\in\mathcal{Y}}\alpha_{\bar{y}}\mathbf{a}_{\bar{y}}=\mathbf{0}.\end{aligned} \tag{25}$$

This gives $\boldsymbol{\alpha}\in\mathcal{P}$ and $\hat{\mathbf{w}}=-A\boldsymbol{\alpha}$, since $A=(\eta\mathbf{a}_{\bar{y}})_{\bar{y}\in\mathcal{Y}}$ by definition. By injecting these constraints in (24), we obtain:

$$\max_{\boldsymbol{\alpha}\in\mathcal{P}}\frac{1}{2\eta}\|A\boldsymbol{\alpha}\|^2+-A\boldsymbol{\alpha}^{\top}\frac{1}{\eta}A\boldsymbol{\alpha}+\mathbf{b}^{\top}\boldsymbol{\alpha}, \tag{26}$$

which finally gives the desired result. □

## A.3 Derivation of the Optimal Step-Size

**Lemma 2** (Optimal Step-Size). *Suppose that we make a step in the direction of* $\mathbf{s} \in \mathcal{P}$ *in the dual. We define the corresponding primal variables* $\mathbf{w_s} = -A\mathbf{s}$ *and* $\lambda_{\mathbf{s}} = \mathbf{b}^\top \mathbf{s}$, *as well as* $\lambda = \mathbf{b}^\top \boldsymbol{\alpha}$. *Then the optimal step-size is given by:*

$$\gamma = \frac{(\mathbf{w} - \mathbf{w}_0 - \mathbf{w_s})^\top (\mathbf{w} - \mathbf{w}_0) + \eta(\lambda_{\mathbf{s}} - \lambda)}{\|\mathbf{w} - \mathbf{w}_0 - \mathbf{w_s}\|^2}. \tag{27}$$

*Proof.* Given the direction $\mathbf{s}$, we take the step $\boldsymbol{\alpha} + \gamma(\mathbf{s} - \boldsymbol{\alpha})$. The new objective is given by:

$$-\frac{1}{2\eta}\|A(\boldsymbol{\alpha} + \gamma(\mathbf{s} - \boldsymbol{\alpha}))\|^2 + \mathbf{b}^\top(\boldsymbol{\alpha} + \gamma(\mathbf{s} - \boldsymbol{\alpha})). \tag{28}$$

In order to compute the optimal step-size, we compute the derivative of the above expression with respect to gamma, and set it to 0:

$$-\frac{1}{\eta}(\mathbf{s} - \boldsymbol{\alpha})^\top A^\top A(\boldsymbol{\alpha} + \gamma(\mathbf{s} - \boldsymbol{\alpha})) + \mathbf{b}^\top(\mathbf{s} - \boldsymbol{\alpha}) = 0. \tag{29}$$

We can isolate the unique term containing $\gamma$:

$$-\frac{1}{\eta}\gamma\|A(\mathbf{s} - \boldsymbol{\alpha})\|^2 - \frac{1}{\eta}(\mathbf{s} - \boldsymbol{\alpha})^\top A^\top A\boldsymbol{\alpha} + \mathbf{b}^\top(\mathbf{s} - \boldsymbol{\alpha}) = 0. \tag{30}$$

This yields:

$$\begin{aligned}
\gamma &= \frac{-\frac{1}{\eta}(\mathbf{s} - \boldsymbol{\alpha})^\top A^\top A\boldsymbol{\alpha} + \mathbf{b}^\top(\mathbf{s} - \boldsymbol{\alpha})}{\frac{1}{\eta}\|A(\mathbf{s} - \boldsymbol{\alpha})\|^2}, \\
&= \frac{-\frac{1}{\eta}(A\mathbf{s} - A\boldsymbol{\alpha})^\top A\boldsymbol{\alpha} + \mathbf{b}^\top(\mathbf{s} - \boldsymbol{\alpha})}{\frac{1}{\eta}\|A\mathbf{s} - A\boldsymbol{\alpha}\|^2}, \\
&= \frac{-(A\mathbf{s} - A\boldsymbol{\alpha})^\top A\boldsymbol{\alpha} + \eta\mathbf{b}^\top(\mathbf{s} - \boldsymbol{\alpha})}{\|A\mathbf{s} - A\boldsymbol{\alpha}\|^2}.
\end{aligned} \tag{31}$$

We can then inject the primal variables and simplify:

$$\begin{aligned}
\gamma &= \frac{(-\mathbf{w_s} + \hat{\mathbf{w}})^\top \hat{\mathbf{w}} + \eta(\lambda_{\mathbf{s}} - \lambda)}{\|-\mathbf{w_s} + \hat{\mathbf{w}}\|^2}, \\
&= \frac{(\mathbf{w} - \mathbf{w}_0 - \mathbf{w_s})^\top(\mathbf{w} - \mathbf{w}_0) + \eta(\lambda_{\mathbf{s}} - \lambda)}{\|\mathbf{w} - \mathbf{w}_0 - \mathbf{w_s}\|^2}.
\end{aligned} \tag{32}$$

$\square$

## A.4 Primal-Dual Proximal Frank-Wolfe Algorithm

We present here the primal-dual algorithm that solves (9) using the previous results:

---
**Algorithm 2** *Proximal Frank Wolfe Algorithm*

---
**Require:** proximal coefficient $\eta$, initial point $\mathbf{w}_0 \in \mathbb{R}^p$, sample $(\mathbf{x}, y)$.
1: $\mathbf{w}_1 = \mathbf{w}_0 - \eta\partial\rho(\mathbf{w})\big|_{\mathbf{w}_0}$  ▷ Initialization $\mathbf{w}_0 - A\boldsymbol{\alpha}$ with $\boldsymbol{\alpha} = \mathbf{1}_y$
2: $\lambda_1 = 0$  ▷ Initialization $\mathbf{b}^\top\boldsymbol{\alpha}$ with $\boldsymbol{\alpha} = \mathbf{1}_y$
3: $t = 1$
4: **while** not converged **do**
5: $\quad$ Choose direction $\mathbf{s}_t \in \mathcal{P}$  ▷ (e.g. conditional gradient or smoothed loss)
6: $\quad \mathbf{w_s} = -A\mathbf{s}_t$
7: $\quad \lambda_{\mathbf{s}} = \mathbf{b}^\top\mathbf{s}_t$
8: $\quad \gamma_t = \dfrac{(\mathbf{w}_t - \mathbf{w}_0 - \mathbf{w_s})^\top(\mathbf{w}_t - \mathbf{w}_0) + \eta(\lambda_{\mathbf{s}} - \lambda_t)}{\|\mathbf{w} - \mathbf{w}_0 - \mathbf{w_s}\|^2}$  ▷ Optimal- step-size
9: $\quad \mathbf{w}_{t+1} = (1 - \gamma_t)\mathbf{w}_t + \gamma_t(\mathbf{w_s} + \mathbf{w}_0)$  ▷ $A\boldsymbol{\alpha}_{t+1} = (1 - \gamma_t)A\boldsymbol{\alpha}_t + \gamma_t A\mathbf{s}_t$
10: $\quad \lambda_{t+1} = (1 - \gamma_t)\lambda_t + \gamma_t\lambda_{\mathbf{s}}$  ▷ $\mathbf{b}^\top\boldsymbol{\alpha}_{t+1} = (1 - \gamma_t)\mathbf{b}^\top\boldsymbol{\alpha}_t + \gamma_t\mathbf{b}^\top\mathbf{s}_t$
11: $\quad t = t + 1$
12: **end while**

---

Note that when $\mathbf{f_x}$ is linear, and when the search direction $\mathbf{s}$ is given by the conditional gradient, we recover the standard Frank-Wolfe algorithm for SVM (Lacoste-Julien et al., 2013).

### A.5 SINGLE-STEP PROXIMAL FRANK-WOLFE ALGORITHM

We now provide some simplification to the steps 6, 8 and 9 of Algorithm 2 when a single step is taken, as is the case in the DFW algorithm. This corresponds to the iteration $t = 1$.

**Proposition 2** (Cost per iteration). *If a single step is performed on the dual of (6), its conditional gradient is given by* $-\partial \left( \rho(\mathbf{w}) + \mathcal{L}_y(\mathbf{f_x}(\mathbf{w})) \right) \big|_{\mathbf{w}_t}$. *The resulting update can be written as:*

$$\mathbf{w}_{t+1} = \mathbf{w}_t - \eta \left[ \partial \rho(\mathbf{w}) \big|_{\mathbf{w}_t} + \gamma \partial \mathcal{L}_j(\mathbf{f}_j(\mathbf{w})) \big|_{\mathbf{w}_t} \right] \tag{33}$$

*Proof.* It is known that for linear SVMs, the direction of the dual conditional gradient is given by the negative sub-gradient of the primal (Lacoste-Julien et al., 2013, Bach, 2015). We apply this result to the Taylor expansion of the network, which is the local model used for the proximal problem. Then we have that at iteration $t = 1$, the conditional gradient is given by:

$$- \partial \left( \mathcal{T}_{\mathbf{w}_0} \rho(\mathbf{w}) + \mathcal{L}_y(\mathcal{T}_{\mathbf{w}_0} \mathbf{f_x}(\mathbf{w})) \right) \big|_{\mathbf{w}_0}. \tag{34}$$

It now suffices to notice that a first-order Taylor expansion does not modify the derivative at its point of linearization: for a function $\phi$, $\partial \mathcal{T}_{\mathbf{w}_0} \phi(\mathbf{w}) \big|_{\mathbf{w}_0} = \partial \phi(\mathbf{w}) \big|_{\mathbf{w}_0}$. By applying this property and the chain rule to (34), we obtain that the conditional gradient is given by:

$$- \partial \left( \rho(\mathbf{w}) + \mathcal{L}_y(\mathbf{f_x}(\mathbf{w})) \right) \big|_{\mathbf{w}_0}. \tag{35}$$

This completes the proof that the conditional gradient direction is given by a stochastic gradient. We now prove equation (33) in the next lemma. □

**Lemma 3.** *Suppose that we apply the Proximal Frank-Wolfe algorithm with a single step. Let* $\boldsymbol{\delta}_t = \partial \left[ \mathbf{s}_t^\top \left( f_{\mathbf{x}, \bar{y}}(\mathbf{w}_0) - f_{\mathbf{x}, y}(\mathbf{w}_0) \right)_{\bar{y} \in \mathcal{Y}} \right]$ *and* $\mathbf{r}_t = \partial_w \rho(\mathbf{w}_0)$. *Then we can rewrite step 6 as:*

$$\mathbf{w}_s = -\eta \left[ \mathbf{r}_t + \boldsymbol{\delta}_t \right]. \tag{36}$$

*In addition, we can simplify steps 8 and 9 of Algorithm 2 to:*

$$\gamma_t = \frac{-\eta \boldsymbol{\delta}_t^\top \mathbf{r}_t + \mathbf{s}_t^\top \mathbf{b}}{\eta \| \boldsymbol{\delta}_t \|^2} \text{ clipped to [0, 1],} \tag{37}$$

$$\mathbf{w}_{t+1} = \mathbf{w}_0 - \eta \left[ \mathbf{r}_t + \gamma_t \boldsymbol{\delta}_t \right]. \tag{38}$$

*Proof.* Again, since we perform a single step of FW, we assume $t = 1$. To prove equation (36), we note that:

$$\begin{aligned} \mathbf{w}_s &= -A\mathbf{s}, \\ &= -\eta \left[ \partial \rho(\mathbf{w}) \big|_{\mathbf{w}_0} + \left( \left( \partial f_{\mathbf{x}, \bar{y}}(\mathbf{w}) \big|_{\mathbf{w}_0} - \partial f_{\mathbf{x}, y}(\mathbf{w}) \big|_{\mathbf{w}_0} \right)_{\bar{y} \in \mathcal{Y}} \right)^\top \mathbf{s}_t \right], \\ &= -\eta \left[ \mathbf{r}_t + \boldsymbol{\delta}_t \right]. \end{aligned} \tag{39}$$

We point out the two following results:

$$\mathbf{w}_t - \mathbf{w}_0 = \mathbf{w}_1 - \mathbf{w}_0 = -\eta \partial \rho(\mathbf{w}) \big|_{\mathbf{w}_0} = -\eta \mathbf{r}_t, \tag{40}$$

and:

$$\mathbf{w}_t - \mathbf{w}_0 - \mathbf{w}_s = -\eta \mathbf{r}_t + \eta \mathbf{r}_t + \eta \boldsymbol{\delta}_t = \eta \boldsymbol{\delta}_t. \tag{41}$$

Since $\lambda_1 = 0$ by definition, equation (37) is obtained with a simple application of equations 40 and 41. Finally, we prove equation 38 by writing:

$$\begin{aligned} \mathbf{w}_{t+1} &= (1 - \gamma_t)\mathbf{w}_t + \gamma_t(\mathbf{w_s} + \mathbf{w}_0), \\ &= (1 - \gamma_t)(\mathbf{w}_0 - \eta \mathbf{r}_t) + \gamma_t(-\eta \mathbf{r}_t - \eta \boldsymbol{\delta}_t + \mathbf{w}_0), \\ &= \mathbf{w}_0 - \eta(\mathbf{r}_t + \gamma_t \boldsymbol{\delta}_t). \end{aligned} \tag{42}$$

□

A.6 SMOOTHING THE LOSS

As pointed out in the paper, the SVM loss is non-smooth and has sparse derivatives, which can prevent the effective training of deep neural networks (Berrada et al., 2018). Partial linearization can solve this problem by locally smoothing the dual (Mohapatra et al., 2016). However, this would introduce a temperature hyper-parameter which is undesirable. Therefore, we note that DFW can be applied with any direction that is feasible in the dual, since it computes an optimal step-size. In particular, the following result states that we can use the well-conditioned and non-sparse gradient of cross-entropy.

**Proposition 3.** *The gradient of cross-entropy in the primal gives a feasible direction in the dual. Furthermore, we can inexpensively detect when this feasible direction cannot provide any improvement in the dual, and automatically switch to the conditional gradient when that is the case.*

For simplicity, we divide Proposition 3 into two distinct parts: first we show how the CE gradient gives a feasible direction in the dual, and then how it can be detected to be an ascent direction.

**Lemma 4.** *The gradient of cross-entropy in the primal gives a feasible direction in the dual. In other words, the gradient of cross-entropy $\boldsymbol{g}$ in the primal is such that there exists a dual search direction $\mathbf{s} \in \mathcal{P}$ verifying $\boldsymbol{g} = -A\mathbf{s}$.*

*Proof.* We consider the vector of scores $(f_{\mathbf{x},\bar{y}}(\mathbf{w}))_{\bar{y} \in \mathcal{Y}} \in \mathbb{R}^{|\mathcal{Y}|}$. We compute its softmax: $\mathbf{s}_{\text{ce}} = \left( \frac{\exp(f_{\mathbf{x},\bar{y}}(\mathbf{w}))}{\sum_{j \in \mathcal{Y}} \exp(f_{\mathbf{x},j}(\mathbf{w}))} \right)_{\bar{y} \in \mathcal{Y}}$. Clearly, $\mathbf{s}_{\text{ce}} \in \mathcal{P}$ by property of the softmax. Furthermore, by going back to the definition of $A$, one can easily verify that $-A\mathbf{s}_{\text{ce}}$ is exactly the primal gradient given by a backward pass through the cross-entropy loss instead of the hinge loss. This concludes the proof. $\square$

The previous lemma has shown that we can use the gradient of cross-entropy as a feasible direction $\mathbf{s}_{\text{ce}}$ in the dual. The next step is to make it a dual ascent direction, that is a direction which always permits improvement on the dual objective (unless at the optimal point). In what follows, we show that we can inexpensively (approximately) compute a sufficient condition for $\mathbf{s}_{\text{ce}}$ to be an ascent direction. If the condition is not satisfied, then we can automatically switch to use the subgradient of the hinge loss (which is known as an ascent direction in the dual).

**Lemma 5.** *Let $\mathbf{s} \in \mathcal{P}$ be a feasible direction in the dual, and $\boldsymbol{v} = (\mathcal{T}_{\mathbf{w}_0}\mathbf{f}_{\mathbf{x}}(\mathbf{w}_t)_{\bar{y}} + \Delta(\bar{y}, y) - \mathcal{T}_{\mathbf{w}_0}\mathbf{f}_{\mathbf{x}}(\mathbf{w}_t)_y)_{\bar{y} \in \mathcal{Y}} \in \mathbb{R}^{|\mathcal{Y}|}$ be the vector of augmented scores output by the linearized model. Let us assume that we apply the single-step Proximal Frank-Wolfe algorithm (that is, we have $t = 1$), and that $\rho$ is a non-negative function.*
*Then $\mathbf{s}^\top \boldsymbol{v} > 0$ is a sufficient condition for $\mathbf{s}$ to be an ascent direction in the dual.*

*Proof.* Let $\mathbf{s} \in \mathcal{P}$, $\boldsymbol{v} = (\mathcal{T}_{\mathbf{w}_0}\mathbf{f}_{\mathbf{x}}(\mathbf{w}_t)_{\bar{y}} + \Delta(\bar{y}, y) - \mathcal{T}_{\mathbf{w}_0}\mathbf{f}_{\mathbf{x}}(\mathbf{w}_t)_y)_{\bar{y} \in \mathcal{Y}}$. By definition, we have that:

$$
\begin{aligned}
\boldsymbol{v} &= \left( \mathbf{a}_{\bar{y}}^\top (\mathbf{w}_t - \mathbf{w}_0) + b_{\bar{y}} - \mathcal{T}_{\mathbf{w}_0}\rho(\mathbf{w}) \right)_{\bar{y} \in \mathcal{Y}}, \\
&= \frac{1}{\eta} A^\top (\mathbf{w}_t - \mathbf{w}_0) + \mathbf{b} - (\mathcal{T}_{\mathbf{w}_0}\rho(\mathbf{w}))_{\bar{y} \in \mathcal{Y}}.
\end{aligned}
\tag{43}
$$

Therefore:

$$
\begin{aligned}
&\mathbf{s}^\top \boldsymbol{v} > 0 \\
\iff & \frac{1}{\eta}(A\mathbf{s})^\top(\mathbf{w}_t - \mathbf{w}_0) + \mathbf{s}^\top \mathbf{b} - \mathbf{s}^\top (\mathcal{T}_{\mathbf{w}_0}\rho(\mathbf{w}))_{\bar{y} \in \mathcal{Y}} > 0, \\
\iff & (A\mathbf{s})^\top(\mathbf{w}_t - \mathbf{w}_0) + \eta\mathbf{s}^\top \mathbf{b} - \eta\mathcal{T}_{\mathbf{w}_0}\rho(\mathbf{w}) > 0, \quad (\text{since } \mathbf{s} \in \mathcal{P} \text{ and } \eta > 0) \\
\iff & -\mathbf{w}_s^\top(\mathbf{w}_t - \mathbf{w}_0) + \eta\mathbf{s}^\top \mathbf{b} - \eta\rho(\mathbf{w}_0) - \eta\partial\rho(\mathbf{w}_0)^\top(\mathbf{w}_t - \mathbf{w}_0) > 0, \\
\iff & -\mathbf{w}_s^\top(\mathbf{w}_t - \mathbf{w}_0) + \eta\mathbf{s}^\top \mathbf{b} - \eta\rho(\mathbf{w}_0) + (\mathbf{w}_t - \mathbf{w}_0)^\top(\mathbf{w}_t - \mathbf{w}_0) > 0, \\
\iff & (\mathbf{w}_t - \mathbf{w}_0 - \mathbf{w}_s)^\top(\mathbf{w}_t - \mathbf{w}_0) + \eta\mathbf{s}^\top \mathbf{b} - \eta\rho(\mathbf{w}_0) > 0, \\
\implies & (\mathbf{w}_t - \mathbf{w}_0 - \mathbf{w}_s)^\top(\mathbf{w}_t - \mathbf{w}_0) + \eta\mathbf{s}^\top \mathbf{b} > 0, \quad (\text{because } \rho(\mathbf{w}_0) \geq 0) \\
\iff & \gamma_t > 0 \quad (\text{we have that } \lambda_t = 0 \text{ at } t = 1).
\end{aligned}
\tag{44}
$$

We have just shown that if $\mathbf{s}^\top \boldsymbol{v} > 0$, then $\gamma_t > 0$. Since $\gamma_t$ is an optimal step-size, this indicates that $\mathbf{s}$ is an ascent direction (we would obtain $\gamma_t = 0$ for a direction $\mathbf{s}$ that cannot provide improvement). $\square$

**Approximate Condition.**    In practice, we consider that $\mathcal{T}_{\mathbf{w}_0} \mathbf{f}_{\mathbf{x}}(\mathbf{w}_t) \simeq \mathbf{f}_{\mathbf{x}}(\mathbf{w}_0)$. Indeed, for $t = 1$, we have that $\|\mathcal{T}_{\mathbf{w}_0} \mathbf{f}_{\mathbf{x}}(\mathbf{w}) - \mathbf{f}_{\mathbf{x}}(\mathbf{w}_0)\| = \mathcal{O}(\|\mathbf{w}_t - \mathbf{w}_0\|)$, and $\|\mathbf{w}_t - \mathbf{w}_0\| = \|\eta \partial_w \rho(\mathbf{w}_0))\|$, which is typically very small (we use a weight decay coefficient in the order of $1e^{-4}$ in our experimental settings).    Therefore, we replace $\mathcal{T}_{\mathbf{w}_0} \mathbf{f}_{\mathbf{x}}(\mathbf{w})$ by $\mathbf{f}_{\mathbf{x}}(\mathbf{w}_0)$ in the above criterion, which becomes inexpensive since $\mathbf{f}_{\mathbf{x}}(\mathbf{w}_0)$ is already computed by the forward pass.

## A.7    NESTEROV MOMENTUM

As can be seen in the previous primal-dual algorithms, taking a step in the dual can be decomposed into two stages: the initialization and the movement along the search direction. The initialization step is not informative about the optimization problem. Therefore, we discard it from the momentum velocity, and only accumulate the step along the conditional gradient (scaled by $\gamma_t \eta$). This results in the following velocity update:

$$\mathbf{z}_{t+1} = \mu \mathbf{z}_t - \eta \gamma_t (\mathbf{r}_t + \boldsymbol{\delta}_t). \tag{45}$$

## B   EXPERIMENTAL DETAILS ON THE CIFAR DATA SETS

### B.1   ADAPTIVE GRADIENT BASELINES: CROSS-VALIDATION (WITHOUT DATA AUGMENTATION)

| $l_2$ | $\eta$ | Accuracy CIFAR-10 (%) | Accuracy CIFAR-100 (%) |
|---|---|---|---|
| 0.0001 | 0.001 | 71.6 | 39.44 |
| **0.0001** | **0.01** | **88.18** | **55.72** |
| 0.0001 | 0.1 | 86.4 | 55.44 |
| 0.0001 | 1 | 68.48 | 20.68 |

Table 4: Cross-Validation for ADAGRAD on DN architecture (best validation accuracy obtained during training).

| $l_2$ | $\eta$ | Accuracy CIFAR-10 (%) | Accuracy CIFAR-100 (%) |
|---|---|---|---|
| 0.0001 | 0.001 | 68.98 | 31.86 |
| **0.0001** | **0.01** | **86.4** | 53.82 |
| 0.0001 | 0.1 | 83.6 | 51.18 |
| 0.0005 | 0.001 | 68.66 | 32.5 |
| **0.0005** | **0.01** | 86.3 | **56.16** |
| 0.0005 | 0.1 | 77.92 | 44.12 |

Table 5: Cross-Validation for ADAGRAD on WRN architecture (best validation accuracy obtained during training).

| $l_2$ | $\eta$ | Accuracy CIFAR-10 (%) | Accuracy CIFAR-100 (%) |
|---|---|---|---|
| 0.0001 | 0.0001 | 86.26 | 50.7 |
| **0.0001** | **0.001** | **89.42** | **63.9** |
| 0.0001 | 0.01 | 81.12 | 51.82 |

Table 6: Cross-Validation for ADAM on DN architecture (best validation accuracy obtained during training).

| $l_2$ | $\eta$ | Accuracy CIFAR-10 (%) | Accuracy CIFAR-100 (%) |
|---|---|---|---|
| 0.0001 | 0.0001 | 79.7 | 41.42 |
| **0.0001** | **0.001** | **86.1** | **58.7** |
| 0.0001 | 0.01 | 80.06 | 50.86 |
| 0.0005 | 0.0001 | 78.88 | 40.08 |
| 0.0005 | 0.001 | 85.14 | 55.26 |
| 0.0005 | 0.01 | 72.54 | 36.82 |

Table 7: Cross-Validation for ADAM on WRN architecture (best validation accuracy obtained during training).

| $l_2$ | $\eta$ | Accuracy CIFAR-10 (%) | Accuracy CIFAR-100 (%) |
|---|---|---|---|
| 0.0001 | 0.0001 | 84.28 | 49.54 |
| **0.0001** | **0.001** | **90.4** | **68.54** |
| 0.0001 | 0.01 | 83.98 | 50.44 |

Table 8: Cross-Validation for AMSGRAD on DN architecture (best validation accuracy obtained during training).

| $l_2$ | $\eta$ | Accuracy CIFAR-10 (%) | Accuracy CIFAR-100 (%) |
|---|---|---|---|
| 0.0001 | 0.0001 | 75.86 | 41.6 |
| **0.0001** | **0.001** | **87.02** | **59.6** |
| 0.0001 | 0.01 | 82.32 | 52.12 |
| 0.0005 | 0.0001 | 75.74 | 42.28 |
| 0.0005 | 0.001 | 86.16 | 57.82 |
| 0.0005 | 0.01 | 75.82 | 36.48 |

Table 9: Cross-Validation for AMSGRAD on WRN architecture (best validation accuracy obtained during training).

| $l_2$ | $\eta$ | Accuracy CIFAR-10 (%) | Accuracy CIFAR-100 (%) |
|---|---|---|---|
| 0.0001 | 0.001 | 72.72 | 40.96 |
| 0.0001 | 0.01 | 83.26 | 53.12 |
| **0.0001** | **0.1** | **91.7** | **59.7** |
| 0.0001 | 1 | 10.16 | 1.16 |

Table 10: Cross-Validation for BPGRAD on DN architecture (best validation accuracy obtained during training).

| $l_2$ | $\eta$ | Accuracy CIFAR-10 (%) | Accuracy CIFAR-100 (%) |
|---|---|---|---|
| 0.0001 | 0.001 | 64.98 | 31.9 |
| 0.0001 | 0.01 | 78.46 | 44.26 |
| **0.0001** | **0.1** | **89.24** | 54.42 |
| 0.0001 | 1 | 16.1 | 1.16 |
| 0.0005 | 0.001 | 68.08 | 33.26 |
| **0.0005** | **0.01** | 85.44 | **59.9** |
| 0.0005 | 0.1 | 88.44 | 51.28 |
| 0.0005 | 1 | 10.16 | 1.16 |

Table 11: Cross-Validation for BPGRAD on WRN architecture (best validation accuracy obtained during training).

## B.2 ADAPTIVE GRADIENT BASELINES: CROSS-VALIDATION (WITH DATA AUGMENTATION)

| $l_2$ | $\eta$ | batchsize | Accuracy CIFAR-10 (%) | Accuracy CIFAR-100 (%) |
|---|---|---|---|---|
| 0.0001 | 0.0001 | 64 | 90.38 | 61.6 |
| 0.0001 | 0.0001 | 128 | 87.86 | 57.82 |
| 0.0001 | 0.0001 | 256 | 86.66 | 53.64 |
| **0.0001** | **0.001** | **64** | 92.52 | **69.66** |
| **0.0001** | **0.001** | **128** | **92.72** | 69.5 |
| 0.0001 | 0.001 | 256 | 92.64 | 67.56 |
| 0.0001 | 0.01 | 64 | 82.1 | 45 |
| 0.0001 | 0.01 | 128 | 83.9 | 53.4 |
| 0.0001 | 0.01 | 256 | 86.86 | 58.1 |

Table 12: Cross-Validation for AMSGRAD on DN architecture with data augmentation (best validation accuracy obtained during training).

| $l_2$ | $\eta$ | batchsize | Accuracy CIFAR-10 (%) | Accuracy CIFAR-100 (%) |
|---|---|---|---|---|
| 0.0001 | 0.0001 | 128 | 90.5 | 64.36 |
| 0.0001 | 0.0001 | 256 | 89.6 | 62.02 |
| 0.0001 | 0.0001 | 512 | 88.26 | 58.68 |
| **0.0001** | **0.001** | **128** | 91.7 | **69.3** |
| 0.0001 | 0.001 | 256 | 91.8 | 68.98 |
| **0.0001** | **0.001** | **512** | **91.88** | 68.64 |
| 0.0001 | 0.01 | 128 | 83.36 | 53.72 |
| 0.0001 | 0.01 | 256 | 84.58 | 57.28 |
| 0.0001 | 0.01 | 512 | 87.42 | 60.68 |
| 0.0005 | 0.0001 | 128 | 91.44 | 65.52 |
| 0.0005 | 0.0001 | 256 | 89.7 | 61.98 |
| 0.0005 | 0.0001 | 512 | 88.48 | 59.1 |
| 0.0005 | 0.001 | 128 | 90.82 | 67.38 |
| 0.0005 | 0.001 | 256 | 91 | 67.58 |
| 0.0005 | 0.001 | 512 | 91.06 | 67.06 |
| 0.0005 | 0.01 | 128 | 72.6 | 34.8 |
| 0.0005 | 0.01 | 256 | 76.56 | 41.82 |
| 0.0005 | 0.01 | 512 | 79.12 | 45.6 |

Table 13: Cross-Validation for AMSGRAD on WRN architecture with data augmentation (best validation accuracy obtained during training).

| $l_2$ | $\eta$ | batchsize | Accuracy CIFAR-10 (%) | Accuracy CIFAR-100 (%) |
|---|---|---|---|---|
| 0.0001 | 0.01 | 64 | 92.8 | 69.12 |
| 0.0001 | 0.01 | 128 | 91.38 | 66.26 |
| 0.0001 | 0.01 | 256 | 89.46 | 60.68 |
| **0.0001** | **0.1** | **64** | **95.34** | 68.04 |
| 0.0001 | 0.1 | 64 | 95.34 | 66.78 |
| 0.0001 | 0.1 | 128 | 94.7 | 73.62 |
| **0.0001** | **0.1** | **128** | 94.7 | **74.1** |
| 0.0001 | 0.1 | 256 | 94 | 70.9 |
| 0.0001 | 1 | 64 | 77.04 | 38.44 |
| 0.0001 | 1 | 128 | 82.56 | 52.12 |
| 0.0001 | 1 | 256 | 87.38 | 60.6 |
| 0.0001 | 10 | 64 | 10.16 | 1.16 |
| 0.0001 | 10 | 128 | 10.16 | 1.16 |
| 0.0001 | 10 | 256 | 10.56 | 1.62 |

Table 14: Cross-Validation for DFW on DN architecture with data augmentation (best validation accuracy obtained during training).

| $l_2$ | $\eta$ | batchsize | Accuracy CIFAR-10 (%) | Accuracy CIFAR-100 (%) |
|---|---|---|---|---|
| 0.0001 | 0.01 | 128 | 93.24 | 71.18 |
| 0.0001 | 0.01 | 256 | 91.8 | 67.36 |
| 0.0001 | 0.01 | 512 | 90.9 | 64.74 |
| 0.0001 | 0.1 | 128 | 94.18 | 74.26 |
| 0.0001 | 0.1 | 256 | 94.66 | 73.24 |
| 0.0001 | 0.1 | 512 | 94.02 | 71.66 |
| 0.0001 | 1 | 128 | 84.7 | 55.1 |
| 0.0001 | 1 | 256 | 89.62 | 61.88 |
| **0.0001** | **1** | **512** | **94.98** | 73.94 |
| 0.0001 | 10 | 128 | 10.72 | 1.32 |
| 0.0001 | 10 | 256 | 10.72 | 4.96 |
| 0.0001 | 10 | 512 | 13.62 | 6.4 |
| 0.0005 | 0.01 | 128 | 94.1 | 72.14 |
| 0.0005 | 0.01 | 256 | 92.72 | 69.96 |
| 0.0005 | 0.01 | 512 | 90.84 | 64.04 |
| 0.0005 | 0.1 | 128 | 88.86 | 63.06 |
| **0.0005** | **0.1** | **256** | 94.68 | **75.34** |
| 0.0005 | 0.1 | 512 | 94.1 | 72.54 |
| 0.0005 | 1 | 128 | 63.3 | 26.9 |
| 0.0005 | 1 | 256 | 72.08 | 38.28 |
| 0.0005 | 1 | 512 | 80.74 | 48.1 |
| 0.0005 | 1 | 512 | 80.74 | 44.52 |
| 0.0005 | 10 | 128 | 10.72 | 1.36 |
| 0.0005 | 10 | 256 | 10.72 | 1.3 |
| 0.0005 | 10 | 512 | 14.18 | 1.98 |

Table 15: Cross-Validation for DFW on WRN architecture with data augmentation (best validation accuracy obtained during training).

### B.3 CONVERGENCE PLOTS

In this section we provide the convergence plots of the different algorithms on the CIFAR data sets without data augmentation. In some cases the training performance can show some oscillations. We emphasize that this is the result of cross-validating the initial learning rate based on the validation set

performance: sometimes a better-behaved convergence would be obtained on the training set with a lower learning rate. However this lower learning rate is not selected because it does not provide the best validation performance.

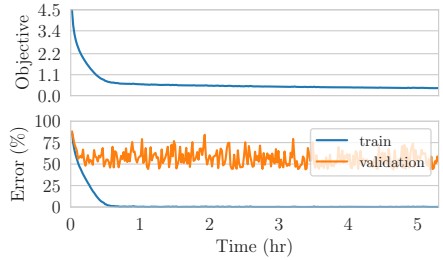

Figure 5: *Convergence plot of Adagrad on CI-FAR 100 with DN architecture.*

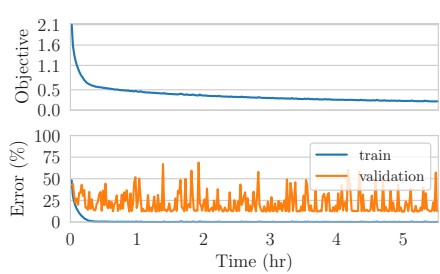

Figure 6: *Convergence plot of Adagrad on CI-FAR 10 with DN architecture.*

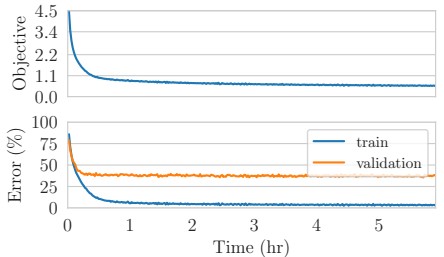

Figure 7: *Convergence plot of Adam on CIFAR 100 with DN architecture.*

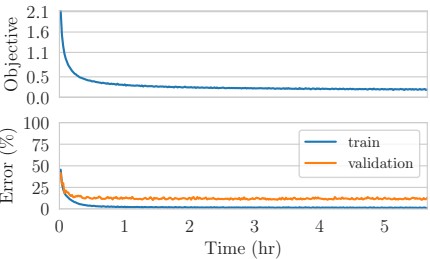

Figure 8: *Convergence plot of Adam on CIFAR 10 with DN architecture.*

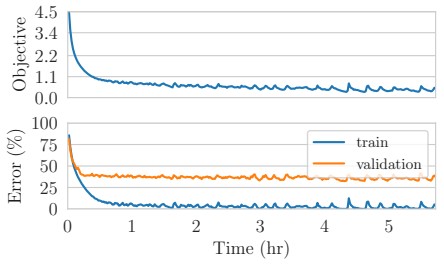

Figure 9: *Convergence plot of AMSGrad on CIFAR 100 with DN architecture.*

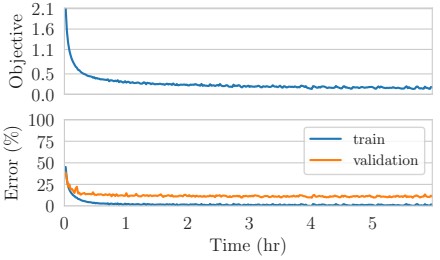

Figure 10: *Convergence plot of AMSGrad on CIFAR 10 with DN architecture.*

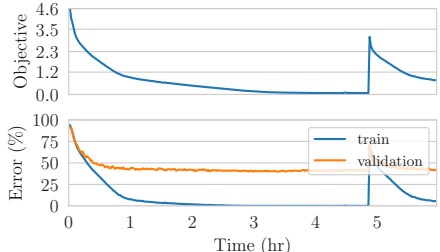

Figure 11: *Convergence plot of BPGrad on CI-FAR 100 with DN architecture.*

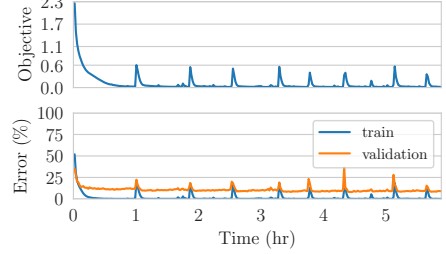

Figure 12: *Convergence plot of BPGrad on CI-FAR 10 with DN architecture.*

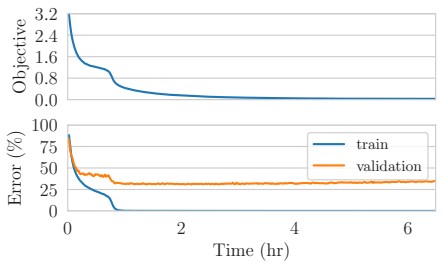

Figure 13: *Convergence plot of DFW on CIFAR 100 with DN architecture.*

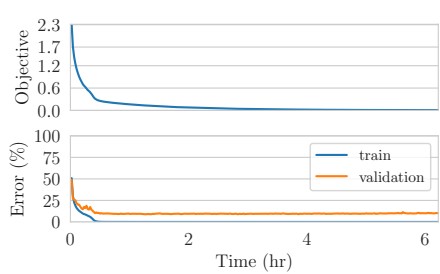

Figure 14: *Convergence plot of DFW on CIFAR 10 with DN architecture.*

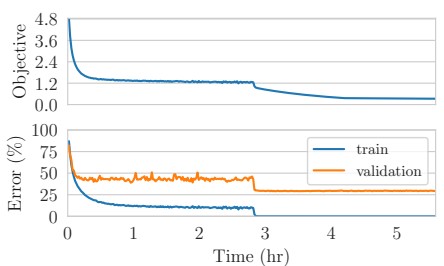

Figure 15: *Convergence plot of SGD on CIFAR 100 with DN architecture.*

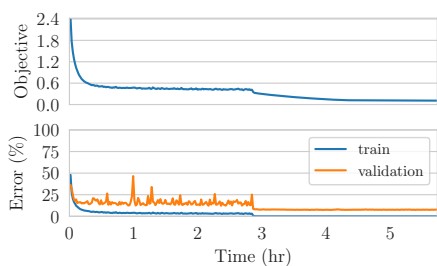

Figure 16: *Convergence plot of SGD on CIFAR 10 with DN architecture.*

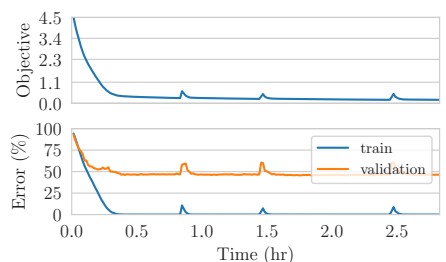

Figure 17: *Convergence plot of Adagrad on CIFAR 100 with WRN architecture.*

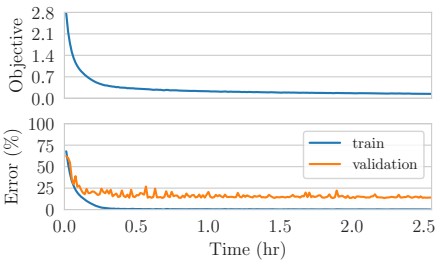

Figure 18: *Convergence plot of Adagrad on CIFAR 10 with WRN architecture.*

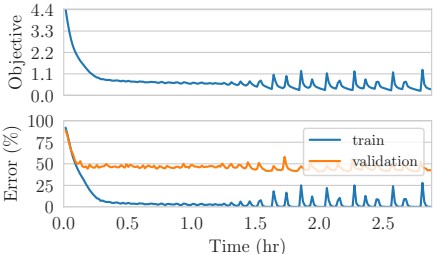

Figure 19: *Convergence plot of Adam on CIFAR 100 with WRN architecture.*

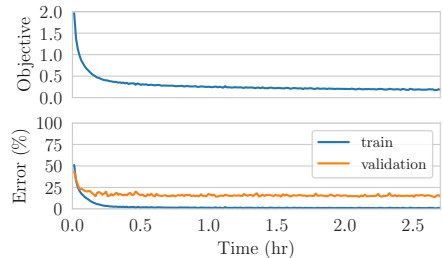

Figure 20: *Convergence plot of Adam on CIFAR 10 with WRN architecture.*

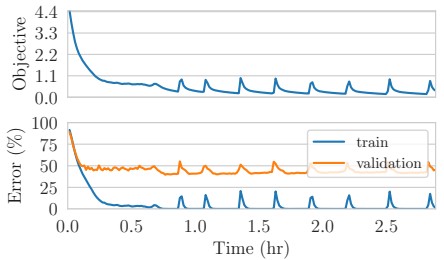

Figure 21: *Convergence plot of AMSGrad on CIFAR 100 with WRN architecture.*

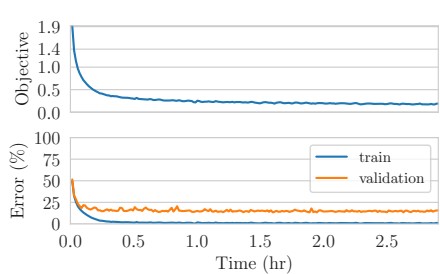

Figure 22: *Convergence plot of AMSGrad on CIFAR 10 with WRN architecture.*

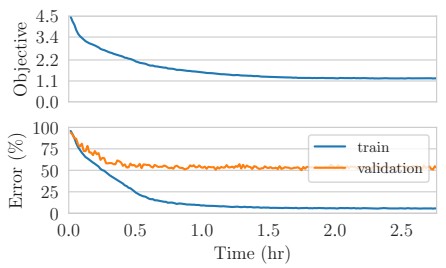

Figure 23: *Convergence plot of BPGrad on CIFAR 100 with WRN architecture.*

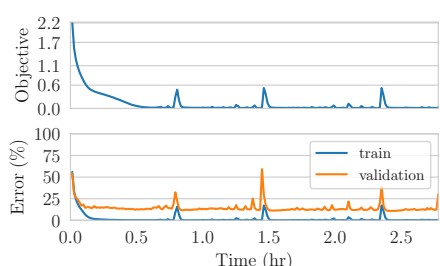

Figure 24: *Convergence plot of BPGrad on CIFAR 10 with WRN architecture.*

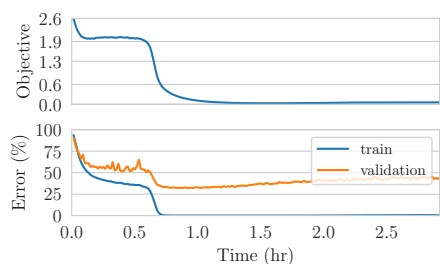

Figure 25: *Convergence plot of DFW on CIFAR 100 with WRN architecture.*

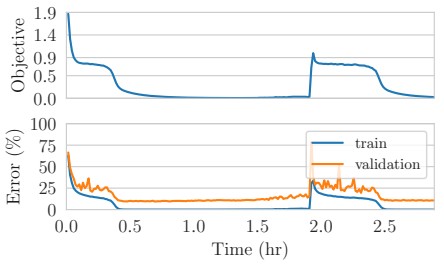

Figure 26: *Convergence plot of DFW on CIFAR 10 with WRN architecture.*

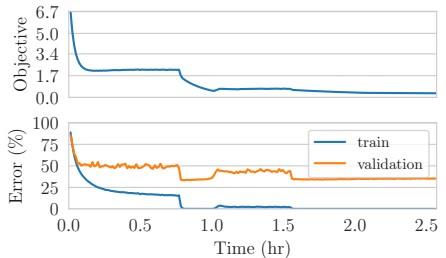

Figure 27: *Convergence plot of SGD on CIFAR 100 with WRN architecture.*

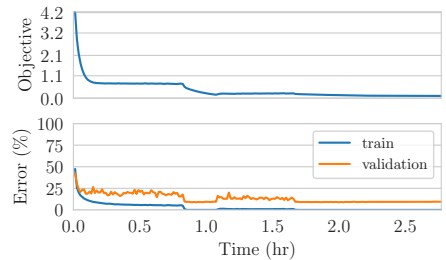

Figure 28: *Convergence plot of SGD on CIFAR 10 with WRN architecture.*

## B.4 SGD & DFW: SENSITIVITY ANALYSIS

We provide here a sensitivity analysis of the DFW algorithm on its hyper-parameter $\eta$, and we compare it against the SGD algorithm with its custom schedule. These experiments do not use data augmentation.

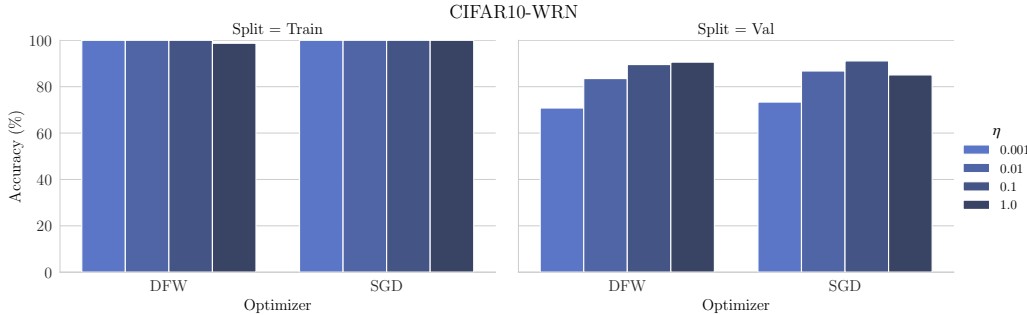

Figure 29: *Sensitivity analysis on the WRN architecture and CIFAR-10 data set.*

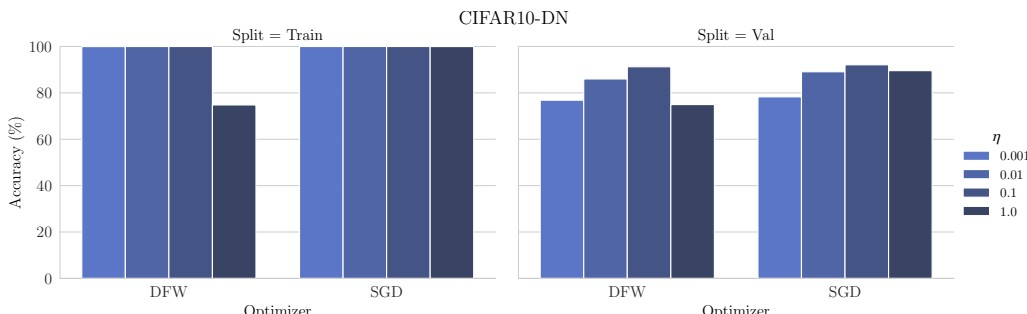

Figure 30: *Sensitivity analysis on the DN architecture and CIFAR-10 data set.*

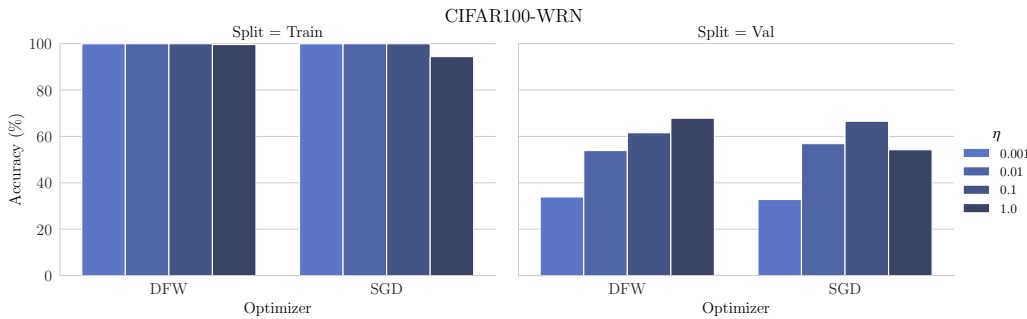

Figure 31: *Sensitivity analysis on the WRN architecture and CIFAR-100 data set.*

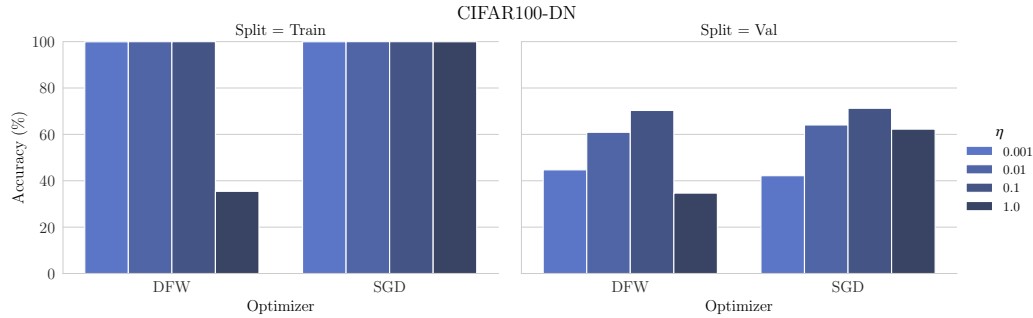

Figure 32: *Sensitivity analysis on the DN architecture and CIFAR-100 data set.*

## C    EXPERIMENTAL DETAILS ON THE SNLI DATA SET

### C.1    CROSS-VALIDATION

| $\eta$ | Accuracy CE (%) | Accuracy SVM (%) |
|---|---|---|
| 0.001 | 83.43 | 84.16 |
| **0.01** | **83.77** | **84.62** |
| 0.1 | 62.09 | 34.5 |

Table 16: Cross-Validation for ADAGRAD on BLSTM architecture (best validation accuracy obtained during training).

| $\eta$ | Accuracy CE (%) | Accuracy SVM (%) |
|---|---|---|
| 1e-05 | 83.18 | 83.02 |
| **0.0001** | **84.56** | **84.69** |
| 0.001 | 84.42 | 83.31 |
| 0.01 | 33.82 | 33.82 |

Table 17: Cross-Validation for ADAM on BLSTM architecture (best validation accuracy obtained during training).

| $\eta$ | Accuracy CE (%) | Accuracy SVM (%) |
|---|---|---|
| 1e-05 | 82.81 | 82.95 |
| **0.0001** | **84.69** | **84.83** |
| 0.001 | 84.66 | 83.59 |
| 0.01 | 36.78 | 38.25 |

Table 18: Cross-Validation for AMSGRAD on BLSTM architecture (best validation accuracy obtained during training).

| $\eta$ | Accuracy CE (%) | Accuracy SVM (%) |
|---|---|---|
| 0.001 | 75.51 | 74.87 |
| 0.01 | 83.09 | 83.02 |
| 0.1 | 83.93 | 84.24 |
| **1.0** | **84.28** | **84.73** |
| 10 | 33.82 | 33.31 |

Table 19: Cross-Validation for BPGRAD on BLSTM architecture (best validation accuracy obtained during training).

| $\eta$ | Accuracy (%) |
|---|---|
| 0.1 | 84.87 |
| **1.0** | **85.21** |
| 10 | 84.76 |

Table 20: Cross-Validation for DFW on BLSTM architecture (best validation accuracy obtained during training).

| $\eta$ | Accuracy CE (%) | Accuracy SVM (%) |
|---|---|---|
| 0.01 | 84.22 | 84.59 |
| **0.1** | 84.63 | **85.15** |
| **1.0** | **85.06** | 84.7 |
| 10 | 34.59 | 34.51 |

Table 21: Cross-Validation for SGD on BLSTM architecture (best validation accuracy obtained during training).

