# OpenReview forum: "Deep Frank-Wolfe For Neural Network Optimization"
_ICLR.cc/2019/Conference_

### Official Review · AnonReviewer3 · 2018-10-31
**Good Paper**

**Rating:** 8
**Confidence:** 4

**Review:**

This paper introduced a proximal approach to optimize neural networks by linearizing the network output instead of the loss function. They demonstrate their algorithm on multi-class hinge loss, where they can show that optimal step size can be computed in close form without significant additional cost. Their experimental results showed competitive performance to SGD/Adam on the same network architectures.

1. Figure 1 is crucial to the algorithm design as it aims to prove that Loss-Preserving Linearization (LPL) preserves information on loss function. While the authors provided numerical plots to compare it with the SGD linearization, I personally prefer to see some analytically comparsion between SGD linearization and LPL even on the simplest case. An appendix with more numerical comparisons on other loss functions might also be insightful.
2. It seems LPL is mainly compared to SGD for convergence (e.g. Fig 2). In Table 2 I saw some optimizers end up with much lower test accuracy. Can the authors show the convergence plots of these methods (similar to Figure 2)?

---

> ### Author Response · Authors · 2018-11-14
> **Thanks for the comments.**
>
> We thank the reviewer for their comments and suggestions. We answer below:
>
> 1. As the reviewer accurately points out, we choose to always employ the hinge loss for DFW in this paper because it gives an optimal step-size. In the new version of the paper, we have included additional baselines on the SNLI data set. This provides more empirical comparisons between the performance of CE and SVM for different optimizers.
>
> 2. In appendix B.2 of the paper, we have added the convergence plot for all methods on the CIFAR data sets.
>
> In some cases the training performance can show some oscillations. We emphasize that this is the result of cross-validating the initial learning rate based on the validation set performance: sometimes a better behaved convergence would be obtained on the training set with a lower learning rate. However this lower learning rate is not selected because it does not provide the best validation performance (this is consistent with our discussion on the step size in section 6).

---

### Official Review · AnonReviewer2 · 2018-11-05
**The proposed DFW lacks of sufficient novelty and the presented performance improvement needs more theoretical justification.**

**Rating:** 7
**Confidence:** 4

**Review:**

This paper proposes a Frank-Wolfe based method, called DFW, for training Deep Network. The DFW method linearizes the loss function into a smooth one, and also adopts Nesterov Momentum to accelerate the training. Both techniques have been widely used in the literature for similar settings. This paper mainly focuses on the algorithm part, but only empirically demonstrate the convergence results.

After reading the authors’ feedback and the paper again, I think overall this is a good paper and should be of broader interest to the broader audience in machine learning community.

In Section 6.1, the authors mention the good generalization is due to large number of steps at a high learning rate. Can we possibly get any theoretical justification on this?

This paper uses multi class hinge loss as an example for illustration. Can this approach be applied for structure prediction, for example, various ranking loss?

---

> ### Author Response · Authors · 2018-11-14
> **Thanks for the comments. Clarification.**
>
> We thank the reviewer for their comments. We provide answers below:
>
> * “The DFW linearizes the loss function into a smooth one, and also adopts Nesterov momentum to accelerate the training.”
> We would like to clarify this statement: one of the key ideas of the DFW algorithm is not to linearize the loss function $\mathcal{L}$, but only the model $f$.
>
> * “Both techniques have been widely used in the literature for similar settings”.
> We wish to clarify the main technical contributions of this paper, since the SVM smoothing and the application of Nesterov acceleration are not the main novelty of this work. We discuss the summary of contributions (available at the end of section 1 of the paper) in the context of technical novelty.
> - Employing a composite framework allows us to use an efficient primal-dual algorithm. As stated by Reviewer 1, this is novel in the context of deep neural networks: “To my knowledge, the submission is the first sound attempt to adapt this type of Dual-based algorithm for optimization of Deep Neural Network [..]”.
> - Crucially, our approach yields an update at the same computational cost per iteration as SGD and with the same level of parallelization. In contrast, in the closest approach to ours, the algorithm of Singh & Shawe-Taylor (2018) can only process a single sample at a time. This results in an approach whose runtime is virtually multiplied by the batch-size (it would be slower by two orders of magnitude in typical classification settings, including for the experiments of this paper).
> - We do not mean to claim that the application of Nesterov acceleration is a technical novelty in itself. However, its use is subtle in our case (see appendix A.7) and it is empirically crucial for good performance, hence its mention in the paper.
> - To the best of our knowledge, the hyper-parameter free smoothing approach that we propose in this work is novel (but is not the main contribution).
>
> We have adapted the abstract and summary of contributions to focus on the main novelty, which is an optimization algorithm for deep neural networks with an optimal step-size at the same computational cost per iteration as SGD.
>
> If the reviewer remains concerned by a lack of novelty, we would be grateful if he/she could provide specific references so that we can compare them in detail with the DFW algorithm.

---

### Official Review · AnonReviewer1 · 2018-11-05
**Interesting approach with room for improvement**

**Rating:** 7
**Confidence:** 5

**Review:**

Dual Block-Coordinate Frank-Wolfe (Dual-BCFW) has been widely used in the literature of non-smooth and strongly-convex stochastic optimization problems, such as (structural) Support Vector Machine. To my knowledge, the submission is the first sound attempt to adapt this type of Dual-based algorithm for optimization of Deep Neural Network, which employs a proximal-point method that linearizes not the whole loss function but only the DNN (up to the logits) to form a convex subproblem and then deal with the loss part in the dual.

The attempt is not perfect (actually with a couple of issues detailed below), but the proposed approach is inspiring and I personally would love it published to encourage more development along this thread.  The following points out a couple of items that could probably help further improve the paper.

*FW vs BCFW*

The algorithm employed in the paper is actually not Frank-Wolfe (FW) but Block-Coordinate Frank-Wolfe (BCFW), as it minimizes w.r.t. a block of dual variables belonging to the min-batch of samples.

*Batch Size*

Though the algorithm can be easily extended to the min-batch case, the author should discuss more how the batch size is interpreted in this case (i.e. minimizing w.r.t. a larger block of dual variables belonging to the batch of samples) and the algorithmic block (Algorithm 1) should be presented in a way reflecting the batch size since this is the way people use an algorithm in practice (to improve the utilization rate of a GPU).

*Convex-Conjugate Loss*

The Dual FW algorithm does not need to be used along with the hinge loss (SVM loss). All convex loss function can derive a dual formulation based on its convex-conjugate. See [1,2] for examples. It would be more insightful to compare SGD vs dual-BCFW when both of them are optimizing the same loss functions (either hinge loss or cross-entropy loss) in the experimental comparison.

[1] Shalev-Shwartz, Shai, and Tong Zhang. "Stochastic dual coordinate ascent methods for regularized loss minimization." JMLR (2013)
[2] Tomioka, Ryota, Taiji Suzuki, and Masashi Sugiyama. "Super-linear convergence of dual augmented Lagrangian algorithm for sparsity regularized estimation." JMLR (2011).

*BCFW vs BCD*

Actually, (Lacoste-Julien, S. et al., 2013) proposes Dual-BCFW to optimize structural SVM because the problem contains exponentially many number of dual variables. For typical multiclass hinge loss problem the Dual Block-Coordinate Descent that minimizes w.r.t. all dual variables of a sample in a closed-form update converges faster without extra computational cost. See the details in, for example, [3, appendix for the multiclass hinge loss case].

[3] Fan, Rong-En, et al. "LIBLINEAR: A library for large linear classification." JMLR (2008).

*Hyper-Parameter*

The proposed dual-BCFW still contains a hyperparameter (eta) due to the need to introduce a convex subproblem, which makes its number of hyperparameters still the same to SGD.

---

> ### Author Response · Authors · 2018-11-14
> **Thanks for the detailed review and the suggestions.**
>
> We thank the reviewer for their detailed review and for their suggestions. We answer point by point:
>
> *FW vs BCFW*
> The (primal) proximal problem is created for a mini-batch of samples, and not for the entire data set (details in section 3.2). In other words, the primal problem consists of the proximal term which encourages proximity to the current iterate, the linearized regularization, and the average over the mini-batch of the losses applied to the linearized model. As a result, we can compute the Frank-Wolfe update for all dual coordinates simultaneously, and we do not need to operate in a block-coordinate fashion. We have included this clarification in the new version of the paper.
>
> *Batch-Size*
> We thank the reviewer for this suggestion. We have adapted the description of Algorithm 1 accordingly.
>
> *Convex-Conjugate Loss*
> In order to compare the DFW algorithm to the strongest possible baselines, we choose the baselines to use the CE loss in the CIFAR experiments. Indeed we have generally found CE to help the baselines in this setting. In addition, the hand-designed learning rate schedule of SGD and the l2 regularization were originally tuned for CE.
> In the case of the SNLI data set, we allow the baseline to use either CE or SVM because using the hinge loss can increase their performance.
> Finally, we choose to always employ the multi-class hinge loss for DFW because it gives an optimal step-size in closed form for the dual, which is a key strength of the formulation.
>
> *BCFW vs BCD*
> We thank the reviewer for this recommendation. It would be interesting indeed to explore how to exploit such updates in the context of the composite minimization framework for deep neural networks. In our case, we emphasize that for speed reasons, it is crucial to process the samples within a mini-batch in parallel, and this does not look straightforward with the algorithm in [3, E.3]. Therefore we believe that for this setting the FW algorithm permits faster updates thanks to an easy parallelization over the mini-batch on GPU.
>
>
> *Hyper-parameter*
> Counting a single hyper-parameter for SGD implicitly assumes that SGD can employ a constant step-size. Using such a constant step-size for SGD would incur a significant loss of performance (e.g. at least a few percents on the CIFAR data set). Therefore in order to obtain good performance, SGD requires a manual schedule of the learning rate, which involves many hyper-parameters to tune in practice.

---

### Public Comment · (anonymous) · 2018-11-26
**Why are your baselines so terrible?**

You applied to the architecture WRN-40-4 to CIFAR10 and CIFAR 100.

As can be seen in Tables 1 and 2, SGD only achieves 90.08 and 66.78 on CIFAR 10 and 100, respectively.

In the original WRN paper (Zagoruyko and Komodakis, 2016 https://arxiv.org/abs/1605.07146 ), the reported results are 95.03 and 77.21 in Table 4. These results are reproducible:

https://github.com/szagoruyko/wide-residual-networks

Similar things also happened to DenseNet. Huang et al 2017 (https://arxiv.org/abs/1608.06993 ) reported 96.54 and 82.82 in Table 2, but yours are 92.02 and 70.33.

Compared with the results in Zagoruyko and Komodakis, 2016, the proposed deep FW algorithm is significantly worse. This is a huge difference!

WRN and DenseNets are two of the most popular architectures, and their good baseline performance is well-known!

---

> ### Author Response · Authors · 2018-11-28
> **The difference is due to data augmentation**
>
> The comment above has pointed out a discrepancy between our results and those from (Zagoruyko and Komodakis, 2016). This is due to the fact that in contrast to (Zagoruyko and Komodakis, 2016), we do not use data augmentation in our CIFAR experiments. Since none of the baselines nor DFW makes use of data augmentation in our experiments, the comparison proposed in this work is valid and fair.
>
> In its current version, the description of our experiments on the CIFAR datasets mistakenly indicates that we use data augmentation, which is not the case. We will correct this in future versions.
>
> As a sanity check, we have verified that our implementation can reproduce the results reported in (Zagoruyko and Komodakis, 2016) when training the model with SGD and with data augmentation.
>
> We will provide results using data augmentation once these are available.

---

> > ### Public Comment · (anonymous) · 2018-11-28
> > **Your baseline is still terrible without data augmentation.**
> >
> > In Table 2 of Huang et al 2017 (https://arxiv.org/abs/1608.06993 ), they reported the accuracies of 94.81 and 80.36 without data augmentation for CIFAR 10 and CIFAR 100, respectively. These are still significantly better than your reported baselines 92.02 and 70.33, especially on CIFAR 100.
> >
> > Furthermore, note that their reported results are only for DenseNet (k=24). For k=40, the results should be even better (very likely to be around 95.XX and 81.XX respectively).

---

> > > ### Author Response · Authors · 2018-11-29
> > > **Our DenseNet architecture has 40 layers and not 250**
> > >
> > > The architecture pointed out in the comment above uses 250 layers. In our experiments, and as specified in section 5.1 of our paper, we use a model with 40 layers. This explains the difference in performance.
> > >
> > > As we have already stated:
> > >
> > > - Since all of the experiments use the same network and same training, the comparison proposed in this work is valid and fair.
> > >
> > > - We have verified that our implementation can reproduce the results reported in (Zagoruyko and Komodakis, 2016) when data augmentation is used, and will provide results using data augmentation once these are available.

---

> > > > ### Author Response · Authors · 2018-12-12
> > > > **Results with data augmentation**
> > > >
> > > > We have performed additional experiments on the CIFAR data sets using data augmentation. We summarise here our findings, and we will provide more details in future versions of the paper.
> > > >
> > > > In order to account for the additional variance introduced by the data augmentation, we allow the batch size to be chosen as 1x, 2x or 4x, where x is the original value of batch-size. Because of the heavy computational cost of the cross-validation (we tune the batch-size, regularization and initial learning rate), we provide results for SGD, DFW and the best performing adaptive gradient method, which is AMSGrad. For SGD the hyper-parameters are kept the same as in (Zagoruyko and Komodakis, 2016) and (Huang et al, 2017), and in particular benefits from hand-designed learning rate schedules. We refer to the Wide Residual Network architecture as WRN, and the DenseNet architecture as DN (details are available in the paper).
> > > >
> > > > We obtain the following results:
> > > > * WRN CIFAR-10  : AMSGrad 90.06, DFW 94.52, SGD 95.40
> > > > * DN CIFAR-10     : AMSGrad 91.78, DFW 94.73, SGD 95.26
> > > > * WRN CIFAR-100: AMSGrad 67.75, DFW 76.12, SGD 77.78
> > > > * DN CIFAR-100    : AMSGrad 69.58, DFW 73.85, SGD 76.26
> > > >
> > > > Comparing these results to Tables 1 and 2, it can be observed that all methods benefit from data augmentation, though with varying increases of performance. DFW systematically and significantly outperforms AMSGrad. In particular, it does so by more than 8% in the WRN-100 case.

---

### Meta-Review · Area_Chair1 · 2018-12-13

**Confidence:** 5
**Recommendation:** Accept (Poster)

**Metareview:**

The paper was judged by the reviewers as providing interesting ideas, well-written and potentially having impact on future research on NN optimization.  The authors are asked to make sure they addressed reviewers comments clearly in the paper.